# In silico analysis of quorum sensing modulators: Insights into molecular docking and dynamics and potential therapeutic applications

Ali Alisaac[ID][1]*

1 Faculty of Applied Medical Sciences, Al-Baha University, Al-Baha, Kingdom of Saudi Arabia

* aalisaac@bu.edu.sa

## Abstract

Quorum sensing (QS) regulates bacterial functions like virulence and biofilm formation, mediated by proteins such as LasI and QscR in *Pseudomonas aeruginosa*. This study investigates the structural dynamics of LasI and QscR proteins in complex with Sulfamerazine and Sulfaperin, using AiiA lactonase as a negative control, through molecular dynamics (MD) simulations to identify potential QS modulators. Molecular docking and MD simulations assessed binding affinity and structural dynamics, analyzing parameters like docking scores, root-mean-square deviation (RMSD), root-mean-square fluctuation (RMSF), solvent-accessible surface area (SASA), radius of gyration (Rg), principal component analysis (PCA), and covariance analysis. Sulfamerazine exhibited the highest binding affinity for LasI based on docking scores, indicating strong ligand-protein interactions. MD simulations revealed stability in the LasI-Sulfamerazine complex, with lower RMSD compared to LasI-Sulfaperin and QscR complexes. RMSF analysis indicated greater flexibility in ligand-binding regions of LasI-Sulfaperin and QscR complexes, suggesting weaker binding. SASA showed a decrease in solvent-accessible surface area for the LasI-Sulfamerazine complex, supporting a compact structure. Rg values confirmed this, with the LasI-Sulfamerazine complex being more compact (~2.00 nm) than QscR-ligand complexes (2.10–2.30 nm). PCA revealed significant conformational changes in the LasI-Sulfamerazine complex, with PC1 explaining 57.26% variance. Covariance analysis indicated stronger residue coupling in the LasI-Sulfamerazine complex, suggesting higher rigidity, while LasI-Sulfaperin and QscR complexes exhibited flexible dynamics. AiiA lactonase was used as a negative control due to its established quorum quenching activity, which hydrolyzes AHL molecules and disrupts QS signaling. Unlike LasI and QscR, AiiA does not rely on small molecule binding for activation. However, a known LasI or QscR inhibitor would have served as a more appropriate positive control, which will be considered in future studies. These findings suggest the LasI-Sulfamerazine complex's stability and rigidity make Sulfamerazine a promising QS modulator. Computational analyses highlight

**Data availability statement:** All relevant data are within the paper and its Supporting Information files.

**Funding:** The author(s) received no specific funding for this work.

**Competing interests:** The authors have declared that no competing interests exist.

its potential to disrupt bacterial communication. Further experimental validation is needed to confirm its therapeutic implications.

---

## 1. Introduction

Antibiotic resistance is widely regarded as one of the most urgent global health crises, placing millions of lives at risk and threatening to undo the medical advancements of the past century [1]. According to the World Health Organization (WHO), antibiotic resistance leads to at least 700,000 deaths annually worldwide, with projections suggesting that, if left unchecked, this number could rise to 10 million by 2050, surpassing cancer as the leading cause of death. The rise of multidrug-resistant (MDR) pathogens is central to this issue, as these bacteria have evolved mechanisms to survive exposure to multiple classes of antibiotics, rendering standard treatments ineffective. This growing resistance is exacerbated by the misuse and overuse of antibiotics in both healthcare and agricultural settings, as well as a slow pace in developing new antibiotics [2,3].

ESKAPE pathogens (*Enterococcus faecium, Staphylococcus aureus, Klebsiella pneumoniae, Acinetobacter baumannii, Pseudomonas aeruginosa,* and *Enterobacter* spp.) represent the leading causes of MDR infections worldwide, with quorum sensing playing a central role in their pathogenicity. While *P. aeruginosa* serves as a model organism for QS studies, similar regulatory systems contribute to virulence and antibiotic resistance in other ESKAPE pathogens. [4,5]. A study by the Centers for Disease Control and Prevention (CDC) estimates that, in the United States alone, over 2.8 million antibiotic-resistant infections occur each year, leading to more than 35,000 deaths. Pathogens such as *Escherichia coli*, *Klebsiella pneumoniae*, *Acinetobacter baumannii*, and *Staphylococcus aureus* are also increasingly resistant to common antibiotics, contributing to a growing healthcare burden and challenging clinicians in their ability to treat infections effectively [6].

Targeting bacterial quorum sensing (QS) has emerged as a promising strategy to mitigate antimicrobial resistance by disrupting coordinated behaviors such as virulence factor production, biofilm formation, and antibiotic resistance. Unlike traditional antibiotics, QS inhibitors (QSIs) interfere with bacterial communication without exerting direct selective pressure, reducing the likelihood of resistance development. [7]. QS is mediated by the production, release, and detection of signaling molecules called acyl-homoserine lactones (AHLs), which accumulate in the bacterial environment as the population density increases [8]. When these molecules reach a threshold concentration, they bind to specific receptor proteins, such as LasR and QscR, activating or repressing the transcription of genes responsible for collective behaviors. Disrupting these signaling pathways can impair bacterial communication, potentially reducing virulence, preventing biofilm formation, and enhancing the efficacy of existing antibiotics [9].

In Gram-negative bacteria such as *P. aeruginosa, E. coli,* and *K. pneumoniae*, QS is predominantly regulated by acyl-homoserine lactones (AHLs) through LasI/LuxI-type homologs. In contrast, Gram-positive bacteria like *Staphylococcus aureus* utilize

an agr-based QS system that relies on autoinducing peptides (AIPs) instead of AHLs. Recognizing these mechanistic differences is crucial for designing broad-spectrum QS inhibitors. However, the QS system is not limited to *P. aeruginosa*—many other bacterial species share similar QS machinery, making this strategy applicable to a broad range of pathogens. For example, in *Escherichia coli* and *Klebsiella pneumoniae*, QS regulates the production of virulence factors such as adhesins, toxins, and enzymes that promote infection and resistance [10,11]. In *Staphylococcus aureus*, the agr system—another QS pathway—regulates the production of exotoxins and biofilm formation, key factors in the pathogenicity of this bacterium [12].

The focus of the current study is to conduct in-silico analyses to identify potential quorum sensing inhibitors (QSIs) that can effectively disrupt the QS pathways of various bacterial species, including *P. aeruginosa*, *E. coli*, *Klebsiella pneumoniae*, and *Staphylococcus aureus* [13]. LasI and QscR regulate key virulence factors in *P. aeruginosa*, including elastase, pyocyanin production, and biofilm formation. These proteins serve as canonical representatives of the LuxI/LuxR quorum sensing system, making them ideal models for studying acyl-homoserine lactone (AHL)-based signaling mechanisms and evaluating potential quorum sensing inhibitors (QSIs). These proteins also share functional similarities with LuxI/LuxR-type systems in *E. coli* and *K. pneumoniae*, suggesting potential for broader application of QSIs. However, the extension of these findings to *S. aureus* would require targeting its agr system, which operates through a distinct autoinducer signaling mechanism., while simultaneously enhancing the activity of conventional antibiotics. In addition, natural enzymes like AiiA lactonase, known for their ability to degrade AHLs and disrupt QS, will be used as a negative control to illustrate the potential benefits of natural QS inhibition [14,15]. This study focuses on AHL-based QS systems due to their prevalence in Gram-negative bacteria such as *P. aeruginosa, E. coli,* and *K. pneumoniae*.

The increasing prevalence of MDR infections and their associated healthcare costs highlight the urgent need for alternative therapeutic strategies. According to a 2019 study in *The Lancet*, the global economic burden of antimicrobial resistance is estimated to reach $100 trillion by 2050 if current trends continue [16]. This underscores the necessity of innovative approaches such as QS inhibition to mitigate the effects of antibiotic resistance. Quorum sensing inhibition has been explored as an alternative to conventional antibiotics, with numerous QSIs identified from natural and synthetic sources. These include synthetic compounds like TZD-C8 and natural compounds such as cinnamaldehyde, ajoene, and baicalin, which interfere with QS-regulated pathways [17–19]. This study builds upon previous efforts by integrating molecular docking and molecular dynamics simulations to assess ligand stability and binding specificity, providing mechanistic insights into QSI interactions with LasI and QscR. These computational analyses pave the way for experimental validation and potential therapeutic development.

## 2. Methods and material

### 2.1. Selection of ligand

The ligand selection process was conducted through a systematic literature review of phytochemicals and synthetic compounds with reported antimicrobial and quorum sensing inhibitory properties. To ensure a rational selection, we first screened 751 compounds from publicly available databases such as Dr. Duke's Phytochemical and Ethnobotanical Databases(https://phytochem.nal.usda.gov/),andPubChem(https://pubchem.ncbi.nlm.nih.gov/).The preliminary screening was based on structural similarity to known QS modulators, documented antimicrobial properties, and drug-like characteristics. After removing redundant and low-relevance compounds, 669 compounds were shortlisted for ADMET analysis. The final set of 23 compounds was selected based on strict ADMET filtering criteria, ensuring optimal pharmacokinetic and safety profiles., and additional details regarding compound selection are provided in Table 2. [20]. Additionally, N-Decanoyl-L-homoserine lactone was included as a known positive control for QscR, and TZD-C8 (Z-5-octylidene-thiazolidine-2,4-dione) was used as a known positive control for LasI[21–23]. PubChem was the sole database used for ligand research, which resulted in the absence of Compound IDs (CIDs) for certain chemicals. After a comprehensive selection process,

 

the chosen compounds were selected for further analysis. The names of these selected compounds were then searched in the PubChem database to retrieve their corresponding three-dimensional structures.

## 2.2. ADMET profiling

ADMET (Absorption, Distribution, Metabolism, Excretion, and Toxicity) profiling was performed using SwissADME (http://www.swissadme.ch/) and pkCSM (https://biosig.lab.uq.edu.au/pkcsm/prediction) tools [24,25]. The canonical SMILES of the selected phytochemicals were compiled into a text file and used as input for both platforms. SwissADME was utilized to evaluate key pharmacokinetic parameters, including absorption, solubility, lipophilicity (LogP), and drug-likeness based on Lipinski's Rule of Five. Lipinski's rule of 5 was applied to filter compounds based on molecular weight, hydrogen bond donors/acceptors, and logP values. In Table 2, a score of 0 indicates no violation, while a higher score reflects deviations from the rule. Compounds with one violation were retained for analysis, while those with multiple violations were excluded to ensure drug-likeness. Additionally, pkCSM predicted toxicity profiles, such as hepatotoxicity, mutagenicity (AMES toxicity), and other toxicological endpoints. Compounds were restricted to a molecular weight range of 150–500 g/mol to optimize absorption and pharmacokinetics. ADMET profiling was performed to identify compounds with optimal absorption, distribution, metabolism, excretion, and toxicity properties, ensuring their suitability for drug development. The ADMET screening parameters and selection thresholds applied in this study are summarized in Table 1. The selection criteria included Lipinski's Rule of Five to assess drug-likeness, where compounds with more than one violation were excluded. Specific thresholds were applied, such as molecular weight between 150−500 g/mol for optimal bioavailability, LogP values between −0.4 to +5.0 to maintain balance between solubility and permeability, and a topological polar surface area (TPSA) under 130 Å² to ensure effective cellular uptake. Toxicity parameters, including hepatotoxicity and mutagenicity (AMES test), were strictly considered, and compounds showing potential toxicity were filtered out. These thresholds were selected based on established guidelines for small-molecule drug discovery to maximize the likelihood of in vivo efficacy. [26]. To ensure the robustness of the findings, several factors were controlled during the study. The protein-ligand complexes were simulated under consistent conditions to mitigate size-related biases. Additionally, ligand selection was based on strict ADMET filtering criteria to minimize variability in drug-like properties.

**Table 1. AMDET profiling parameters and shorting criteria.**

| Parameter | Range/criteria |
|---|---|
| Molecular weight (g/mol) | 150–500 |
| Rotatable bond count | 0–10 |
| Heavy atom count | 20–70 |
| H-bond donor count | 0–5 |
| H-bond acceptor count | 0–10 |
| Topological polar surface area (å²) | 20–130 |
| Complexity | 0–1000 |
| Xlogp | −0.4 to +5.0 |
| Blood-brain barrier (bbb) permeability | Preferred for cns-active drugs; limited permeability for non-cns drugs |
| Lipinski's rule of five | No more than one violation: |
|  | - molecular weight ≤ 500 g/mol |
|  | - log p ≤ 5 |
|  | - h-bond donors ≤ 5 |
|  | - h-bond acceptors ≤ 10 |
| Hepatotoxicity | Absence preferred; assess using in silico predictions and in vitro assays |
| Ames test (mutagenicity) | Negative result preferred; assess using in silico predictions and in vitro assays |

## 2.3. Protein selection

The crystal structures of quorum sensing proteins QscR (PDB ID: 6CC0) [27] and LasI (PDB ID: 1RO5) [28] both derived from *Pseudomonas aeruginosa*, and AiiA lactonase (PDB ID: 7L5F) [29], derived from *Bacillus thuringiensis*, were retrieved from the RCSB Protein Data Bank. (https://www.rcsb.org). These structures, originating from Pseudomonas aeruginosa and Bacillus thuringiensis, provide critical insights into quorum sensing regulation and disruption. Structural data were used for docking studies to identify potential inhibitors targeting QscR and LasI, with AiiA serving as a natural quorum sensing disruptor for control comparison [30].

## 2.4. Molecular docking

The compounds that successfully passed ADMET profiling were selected for molecular docking studies using PyRx 0.8 (https://pyrx.sourceforge.io). Ligands were energy-minimized within the PyRx suite using the steepest descent algorithm and the Universal Force Field (UFF) [31]. The target protein was prepared by converting it into pdbqt format, and the grid box size was adjusted to encompass the entire protein surface, ensuring unbiased exploration of potential binding sites. Docking was performed in triplicate to ensure reproducibility, with results expressed in kcal/mol. More negative docking scores corresponded to stronger binding affinities. Binding poses were examined for interactions with key residues and cross-referenced with the co-crystal ligand to validate the binding within the active site.

The docking results were visualized using PyMol and Biovia Discovery Studio Visualizer 2021 [32]. Both the output. pdbqt file and the prepared macromolecule were loaded simultaneously in PyMol. From the nine conformations generated during docking, only those with a root mean square deviation (RMSD) of 0 were selected for detailed analysis. Binding affinities of the ligand interactions were documented in a tabular format. The resulting protein-ligand complexes were saved in pdb format for further analysis, focusing on identifying key binding sites.

## 2.5. Molecular dynamic simulation

A 200 ns molecular dynamics (MD) simulation was conducted using GROMACS 2023 to explore the system's dynamic behavior under physiological conditions. The TIP3P modified water model was employed to accurately simulate the solvent environment, while the CHARMM36 all-atom force field, as described by Huang and MacKerell Jr. (2024) [33], was used for both the protein and the ligands. To ensure electrical neutrality, Na+ and Cl- ions were added. Energy minimization was carried out before the simulation to resolve any steric clashes and optimize the system's geometry, providing a stable starting configuration. The system was then equilibrated in two phases: first at constant volume and temperature (NVT) at 300 K, followed by equilibration at constant pressure and temperature (NPT) at 1 bar, allowing for volume adjustments [34].

Several analyses were performed to interpret the MD simulation data. The Root Mean Square Deviation (RMSD) was calculated to assess the overall stability of the protein structure and verify the achievement of equilibrium, with the 'Lig fit Protein' parameter focusing on the interaction between the ligand and the protein. Root Mean Square Fluctuation (RMSF) was utilized to evaluate the flexibility of specific protein regions by measuring atomic deviations over time. The Radius of Gyration (Rg) was analyzed to examine the compactness of the protein throughout the simulation, representing the average distance between the protein's center of mass and its extremities. Solvent Accessible Surface Area (SASA) calculations were conducted to gain insights into protein folding and stability by measuring the solvent-exposed surface area. Additionally, Covariance analysis was utilized to determine the correlated motions between residues in the protein-ligand complexes. To further elucidate the connectivity between residues, mutual information analysis could be employed in future studies to quantify dependencies and reveal functionally linked regions within the protein. The simulation results were visualized using the matplotlib package in Python to create detailed graphical representations of the analyzed parameters.

## 2.6. Re-Simulation and blind protein–ligand docking by replica-exchange monte carlo simulation

In this study, a re-simulation of the protein-ligand complex was conducted to assess the stability and conformational changes after a 200 ns molecular dynamics (MD) simulation. The final complexed structure obtained from this simulation was then used as the input for blind protein–ligand docking using EDock (https://zhanggroup.org/EDock/), a replica-exchange Monte Carlo simulation-based web server [35]. The EDock approach was employed to predict potential binding sites and evaluate the interactions between the protein and ligand. By performing the docking simulation with the re-simulated complex, we aimed to gain a deeper understanding of the binding mechanism and to identify potential ligand candidates for further experimental validation. The results provided critical insights into the binding affinity and conformation of the ligand within the active site of the protein, supporting the identification of promising molecules for drug development

## 2.7. Cluster analysis of RMSD using GROMACS

Cluster analysis was conducted to investigate the conformational dynamics of the system over the course of the molecular dynamics (MD) simulation. The GROMACS software package was employed to perform RMSD-based clustering using the gmx cluster command. Input files, including the trajectory file (traj.xtc), topology file (topol.tpr), and index file (index.ndx), were used to define the atom groups for clustering. The trajectory was pre-processed to remove periodic boundary conditions and center the protein using the gmx trjconv command. The RMSD-based clustering was performed with a 0.2 nm cutoff, using the GROMOS method to identify distinct conformational states. The resulting clusters were visualized using an XPM file (clusters.xpm), which was further analyzed with GIMP to inspect the distribution and characteristics of the clusters. The cluster.xvg file, which provides the RMSD for each cluster over time, was plotted to evaluate cluster stability, while the cluster.xtc file, containing the centroids of the clusters, allowed for structural comparisons of the dominant conformations. Additionally, further analysis was performed by comparing the RMSD of the cluster centroids to a reference structure using the gmx rms command. This cluster analysis provided a comprehensive understanding of the structural dynamics, revealing both stable and transient conformational states during the simulation [36].

## 3. Results

The ligand selection process identified a total of 669 unique compounds with antibacterial and antimicrobial potential. These included both phytochemical and chemical compounds derived from literature and database research. The final list was refined from an initial pool of 751 compounds by removing duplicates and validating the biological relevance of each compound.

### 3.1. ADMET analysis

After screening 669 phytochemicals, an ADMET (Absorption, Distribution, Metabolism, Excretion, and Toxicity) profiling was conducted to filter candidates based on specific threshold criteria. Initially, 669 phytochemicals were evaluated for ADME properties, with most failing to meet the necessary criteria for further analysis. Only 98 of these compounds passed the ADME thresholds and were selected for toxicity testing. All ligands screened, along with their ADMET properties, are provided in the supplemental file (Supplementary CSV 1). All screened ligands and their ADMET profiles are provided here. The selection of phytochemicals was based on a range of pharmacokinetic and toxicity parameters to identify compounds with desirable properties. Phytochemicals featuring more than five hydrogen bond donors or over ten hydrogen bond acceptors were excluded, as these characteristics can impair absorption and permeation. To optimize absorption, compounds were restricted to a molecular weight of 500 g/mol or less. Toxicity testing required negative results for both AMES toxicity and hepatotoxicity. Additional criteria included the evaluation of polar surface area and the number of rotatable bonds. Phytochemicals with a polar surface area of 145 Å² or less and no more than 10 rotatable bonds were considered more likely to have favorable oral bioavailability, as an excess of rotatable bonds may reduce permeation efficiency.

Moreover, the correlation between molecular weight and heavy atom count was analyzed, with an ideal heavy atom count of around 36, corresponding to a molecular weight near g/mol. These carefully chosen parameters ensured that the selected phytochemicals possessed optimal pharmacokinetic profiles, low toxicity, and good potential for oral bioavailability, making them suitable candidates for further exploration in drug development. (Table 2). These criteria ensured that the selected phytochemicals possessed favorable pharmacokinetic properties, minimal toxicity, and strong potential for oral bioavailability, making them promising candidates for drug discovery and development.

### 3.2. Molecular interaction at the active site

The ADMET-filtered 23 phytochemicals were analyzed through docking in PyRx (Table 3). AiiA lactonase was included as a reference quorum quenching enzyme, while no specific positive control was used since no established LasI or QscR inhibitors were available for comparison.

Notably, the positive control **TZD-C8** ((z)-5-Octylidenethiazolidine-2,4-dione) showed a docking score of **–7.4 kcal/mol** for LasI, placing it within the effective binding range compared to other ligands, while **N-dodecanoyl-L-homoserine lactone**, a well-established QscR modulator, exhibited a binding score of **–7.5 kcal/mol**. While a positive control with both LasI and QscR inhibitor was not included as the known drug for both QscR and LasI have not yet discovered. While AiiA

**Table 2. The ADMET analysis of the selected compounds.**

| S.N | CID | Compound Name | GI absorption | BBB permeant | AMES toxicity | Hepato toxicity | Lipinki rules 5 violation | Skin Sensitisation |
|---|---|---|---|---|---|---|---|---|
| 1 | 3552 | Halazone | High | No | No | No | 0 | No |
| 2 | 4421 | Nalidixic Acid | High | No | No | No | 0 | No |
| 3 | 4735 | Pentamidine | High | No | No | No | 0 | No |
| 4 | 5319 | Sulfabenzamide | High | No | No | No | 0 | No |
| 5 | 5325 | Sulfamerazine | High | No | No | No | 0 | No |
| 6 | 5327 | Sulfamethazine | High | No | No | No | 0 | No |
| 7 | 5336 | Sulfapyridine | High | No | No | No | 0 | No |
| 8 | 5344 | Sulfisoxazole | High | No | No | No | 0 | No |
| 9 | 7322 | 2-Hydroxy-5-sulfobenzoic acid | High | No | No | No | 0 | No |
| 10 | 8281 | Sulfadicramide | High | No | No | No | 0 | No |
| 11 | 11974 | Dibrompropamidine | High | No | No | No | 0 | No |
| 12 | 64949 | Propamidine | High | No | No | No | 0 | No |
| 13 | 68760 | Brodimoprim | High | No | No | No | 0 | No |
| 14 | 68780 | Protiofate | High | No | No | No | 0 | No |
| 15 | 68933 | Sulfaperin | High | No | No | No | 0 | No |
| 16 | 893742 | 5-chloro-N-(4-fluorobenzyl)thiophene-2-sulfonamide | High | No | No | No | 0 | No |
| 17 | 2353996 | 5-chloro-N-(furan-2-ylmethyl)thiophene-2-sulfonamide | High | No | No | No | 0 | No |
| 18 | 2796468 | N-(carbamoylcarbamothioyl)-2-chlorobenzamide | High | No | No | No | 0 | No |
| 19 | 5280343 | Quercetin | High | No | No | No | 0 | No |
| 20 | 9909368 | Ginkgolide-a | High | No | No | No | 0 | No |
| 21 | 28025864 | Antimicrobial TH-8 | High | No | No | No | 0 | No |
| 22 | 50930787 | 3-(4-hydroxy-3,5-dimethylcyclohexyl)-N-(2-hydroxyethyl)propanamide | High | No | No | No | 0 | No |
| 23 | 108886377 | Antimicrobial agent-3 | High | No | No | No | 0 | No |

**Table 3. The binding affinity of the selected compounds (Kcal/mol) and their interacting residues.**

| SN | CID | Compound Name | Binding Affinity (LasI) | Binding Affinity (QscR) | Binding Affinity (AiiA lactonase) |
|---|---|---|---|---|---|
| 1 | 46220260 | (z)-5-Octylidenethiazolidine-2,4-dione (LasI Inhibitor) | −7.4±0.01 | N/A | −7.5±0.05 |
| 2 | 10221437 | N-dodecanoyl-L-Homoserine lactone | N/A | −7.5±0.23 | −7.1±0.15 |
| 3 | 9909368 | Ginkgolide-a | −8.2±0.01 | −7.1±0.015 | −7.8±0.5 |
| 4 | 68933 | Sulfaperin | −7.9±0.02 | −9.1±0.01 | −7.1±0.015 |
| 5 | 5280343 | Quercetin | −7.9±0.01 | −6.9±0.02 | −7.4±0.02 |
| 6 | 5325 | Sulfamerazine | −7.8±0 | −8.7±0.023 | −7.1±0.25 |
| 7 | 5344 | Sulfisoxazole | −7.5±0 | −8.6±0.23 | −8.4±0.25 |
| 8 | 108886377 | Antimicrobial agent-3 | −7.5±0.01 | −8±0.01 | −8.3±0.015 |
| 9 | 5336 | Sulfapyridine | −7.5±0.02 | −7.3±0.0 | −7.5±0.025 |
| 10 | 5327 | Sulfamethazine | −7.5±0.04 | −6.3±0.0 | −7.7±0.02 |
| 11 | 8281 | Sulfadicramide | −7.5±0.03 | −5.2±0.012 | −7.5±0.0 |
| 12 | 893742 | 5-chloro-N-(4-fluorobenzyl)thiophene-2-sulfonamide | −7.4±0.06 | −9.1±0.015 | −7.5±0.01 |
| 13 | 50930787 | 3-(4-hydroxy-3,5-dimethylcyclohexyl)-N-(2-hydroxyethyl)propanamide | −7.4±0.35 | −5.4±0.01 | −6.7±0.0 |
| 14 | 5319 | Sulfabenzamide | −7.3±0.45 | −6.4±0.034 | −8.4±0.025 |
| 15 | 4421 | Nalidixic Acid | −7.3±0.03 | −5.7±0.25 | −7±0.0 |
| 16 | 2353996 | 5-chloro-N-(furan-2-ylmethyl)thiophene-2-sulfonamide | −7.2±0.01 | −8.1±0.01 | −7.4±0.01 |
| 17 | 28025864 | Antimicrobial TH-8 | −7.2±0.24 | −5.9±0.2 | −8.6±0.0 |
| 18 | 3552 | Halazone | −7±0 | −6.9±0.34 | −6.8±0.015 |
| 19 | 2796468 | N-(carbamoylcarbamothioyl)-2-chlorobenzamide | −6.7±0 | −8.9±0.2 | −7.4±0.01 |
| 20 | 4735 | Pentamidine | −6.7±0.01 | −6.1±0.25 | −7.4±0.2 |
| 21 | 68780 | Protiofate | −6.7±0.02 | −4.9±0.35 | −6.3±0.25 |
| 22 | 7322 | 2-Hydroxy-5-sulfobenzoic acid | −6.5±0.01 | −6.4±0.25 | −6.2±0.20 |
| 23 | 68760 | Brodimoprim | −6.5±0.01 | −5.6±0.0 | −6.9±0.01 |
| 24 | 11974 | Dibrompropamidine | −5.7±0 | −6.8±0.01 | −7.7±0.00 |
| 25 | 64949 | Propamidine | −5.7±0 | −6.8±0.0 | −7.5 |

Sulfaperin (CID 68933) and Sulfamerazine (CID 5325) interact with conserved binding motifs in both LasI and QscR, suggesting favorable binding stability across these QS proteins.

lactonase was included in the docking analysis as a reference quorum quenching enzyme, it was not subjected to molecular dynamics (MD) simulations. AiiA served as a benchmark for assessing ligand binding specificity to LasI and QscR, given its established quorum quenching function. However, its interactions with ligands or QS proteins were not modeled in the MD simulations. Future studies could include MD simulations of AiiA-ligand complexes to further evaluate its role as a negative control. The binding activity was assessed to identify ligands with strong interaction potential. (Table 3). The comparative analysis of the binding affinities of various compounds to LasI, QscR, and AiiA lactonase reveals notable differences in specificity. Ginkgolide-a exhibited moderate binding to LasI (−8.2 kcal/mol), slightly stronger than its interaction with AiiA (−7.8 kcal/mol), suggesting a preferential, though not strong, affinity for LasI. Sulfaperin showed a pronounced preference for QscR (−9.1 kcal/mol), significantly stronger than both LasI (−7.9 kcal/mol) and AiiA (−7.1 kcal/mol), indicating it as a more effective QscR binder. In contrast, Quercetin and Sulfamerazine, while binding moderately to LasI (−7.9 and −7.8 kcal/mol, respectively), did not demonstrate strong selectivity when compared to AiiA (−7.4 and −7.1 kcal/mol), suggesting their affinity for the target proteins is less pronounced. Overall, while many compounds display higher binding affinity to quorum sensing proteins (LasI and QscR) compared to AiiA lactonase, some show weaker interactions with the target proteins, highlighting the need for further optimization for higher specificity and potency. The top 4 ligands protein complex were selected for simulation based on their binding affinity. Table 3 shows binding energy of compounds.

The interactions between the LasI and QscR proteins with their respective ligands (Sulfamerazine, Sulfaperin, Quercetin, Ginkgolide A, and 5-chloro-N-(4-fluorobenzyl) thiophene-2-sulfonamide) are characterized by a variety of stabilizing forces, including hydrogen bonds, electrostatic interactions, and van der Waals forces. In LasI-Sulfamerazine (CID 5325), GLN A:189 forms a conventional hydrogen bond with the ligand's oxygen, while CYS A:199 forms a pi-sulfur interaction, stabilizing the complex. In LasI-Sulfaperin (CID 68933), ARG A:220 establishes another hydrogen bond, and CYS A:199 continues to engage in pi-sulfur interactions. In LasI-Quercetin (CID 5280343), multiple hydrogen bonds occur between GLN A:189 and the ligand, with CYS A:199 involved in pi-donor hydrogen bonds in addition to pi-sulfur interactions, highlighting its versatile role in stabilizing the complex. In LasI-Ginkgolide A (CID 9909368), CYS A:199 again participates in both types of interactions, ensuring binding stability. Moving to QscR, in QscR-Sulfamerazine (CID 5325), the same key residues are involved, with CYS A:199 forming pi-sulfur interactions. In QscR-Sulfaperin (CID 68933), CYS A:199 plays a dual role, engaging in both pi-sulfur and pi-donor hydrogen bonds. In QscR-5-chloro-N-(4-fluorobenzyl) thiophene-2-sulfonamide (CID 893742), CYS A:199 forms both pi-sulfur and pi-donor hydrogen bonds, stabilizing the complex and enhancing binding specificity. Finally, in QscR with N-(carbamoylcarbamothioyl)-2-chlorobenzamide (CID 2796468), CYS A:199 once again participates in pi-sulfur and pi-donor hydrogen bonds, showcasing its critical role in stabilizing the ligand and supporting the binding modes of different ligands within the active site. While van der Waals interactions, though weaker, are observed across all complexes, they contribute to the fine-tuning of the ligand's positioning within the binding site. These interactions highlight the versatility and importance of CYS A:199, which forms multiple types of stabilizing interactions, enhancing the binding affinity and specificity of the ligands in both LasI and QscR complexes (Fig 1).

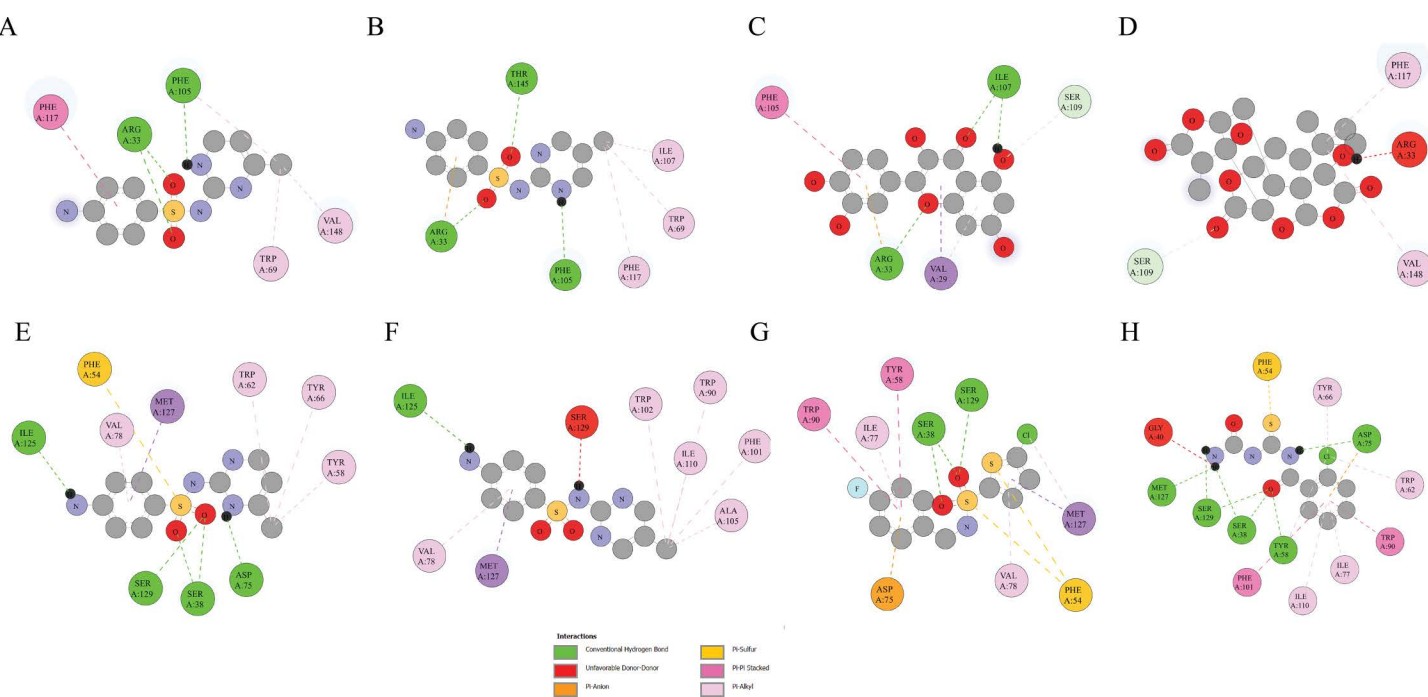

**Fig 1. Two-dimensional interactions with the selected compounds with LasI and QscR.** (A) LasI (PDB ID 1RO5) complexed with Sulfamerazine (CID 5325), illustrating the binding mode and interaction sites. (B) LasI complexed with Sulfaperin (CID 68933), showing the conformational arrangement of the ligand within the binding pocket. (C) LasI complexed with Quercetin (CID 5280343), highlighting the interaction network between the protein and ligand. (D) LasI complexed with Ginkgolide A (Ligand ID 9909368), depicting the spatial orientation of the ligand within the protein's active site. (E) QscR (PDB ID 6CC0) complexed with Sulfamerazine (CID 5325), emphasizing the key residues involved in ligand binding. (F) QscR complexed with Sulfaperin (CID 68933), showing the ligand-protein interaction pattern. (G) QscR complexed with 5-chloro-N-(4-fluorobenzyl) thiophene-2-sulfonamide (CID 893742), revealing the detailed binding interactions. (H) QscR complexed with N-(carbamoylcarbamothioyl)-2-chlorobenzamide (CID 2796468), showcasing the dual binding modes of the ligands in the protein's active site.

## 3.3. Molecular dynamics simulation analysis

The molecular dynamics (MD) simulations were conducted for 200 nanoseconds to ensure sufficient sampling of ligand-protein interactions. While 100-ns simulations are common, extending the duration to 200 ns provided a more comprehensive assessment of complex stability, allowing us to capture long-term conformational changes and fluctuations. This extended timeframe enhances reliability by reducing the influence of transient interactions and providing a more robust evaluation of ligand binding. The RMSD and RMSF analyses showed that complexes stabilized after 150 ns, further supporting the choice of simulation duration. Future studies may explore even longer simulations or enhanced sampling techniques such as replica-exchange molecular dynamics (REMD) for deeper insights into ligand behavior. To assess the statistical reliability of docking results, all docking simulations were performed in triplicate, and the mean binding energy with standard deviation was reported for each ligand. Additionally, non-parametric statistical tests, such as the Mann-Whitney U test, were applied to compare binding affinities between high- and low-scoring ligands.

For MD simulation metrics, the normality of RMSD and RMSF distributions was assessed using the Shapiro-Wilk test, and variance differences were tested with the Levene's test. Principal component analysis (PCA) was conducted to quantify dominant motion patterns in protein-ligand complexes, ensuring that observed conformational changes were statistically significant. These statistical validations strengthen the robustness of our computational findings. The Mann–Whitney U test result is, U statistic = 104.0 p-value = 0.800. Since the p-value is much greater than 0.05, there is no statistically significant difference between the binding affinities of the compounds toward LasI and QscR proteins (S1 Fig).

Only the complexes that exhibited favorable results for these parameters were included for further analysis of the binding grooves. The selected complexes included LasI-Sulfamerazine, LasI-Sulfaperin, QscR-Sulfamerazine, and QscR-Sulfaperin. These stable complexes displayed deep binding grooves with shared residues involved in interactions. Additionally, the residues participating in these interactions showed notable consistency across all four ligands. This congruency in binding groove structures and interacting residues suggests a unified mechanism of binding, supporting the likelihood of a common molecular process or shared target interaction. The pocket region of the complexes (Figs 2 and 3) highlights the ligands bound in the same site, with residues showing similar patterns of interaction. Furthermore, the results also indicate that despite minor variations in ligand size and functional groups, the commonality in binding residues across the complexes suggests a high degree of specificity and suggests that these ligands may share a similar mechanism of action, interacting with LasI and QscR through analogous pathways.

The analysis of the protein-ligand complexes over 200 ns reveals varying degrees of stability based on RMSD values. LasI complexed with Sulfamerazine (CID 5325) (Fig 4A) exhibits significant fluctuations in both the protein and ligand RMSD. In between 100–140 ns the protein and ligand shown a distance and suggesting an unstable interaction and

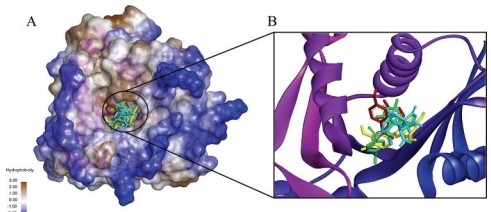

**Fig 2. A 3D interactions shows the binding of four ligands to LasI (PDB ID 1RO5):** Sulfamerazine (CID 5325) (Red), Sulfaperin (CID 68933) (Green), Quercetin (CID 5280343) (Yellow), and Ginkgolide A (CID 9909368) (Cyan). The binding grooves of all ligands were found to share a similar spatial arrangement, with most binding sites exhibiting a predominantly hydrophobic nature. Sulfamerazine and Sulfaperin (Red and Green) show deeper binding grooves, which facilitate stronger interactions, while Quercetin and Ginkgolide A (Yellow and Cyan) display a more balanced distribution of hydrophobic and hydrophilic residues. Despite this variation, all complexes share common interaction residues, suggesting a consistent binding mode across the ligands and supporting the idea of a shared mechanism of interaction with LasI.

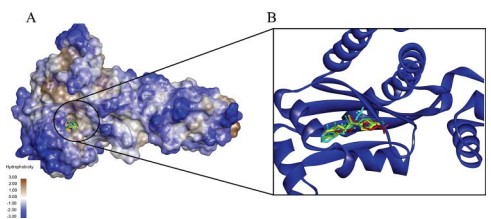

**Fig 3. The binding of four ligands to QscR (PDB ID 6CC0):** Sulfamerazine (CID 5325) (Red), Sulfaperin (CID 68933) (Green), Chloro-N-(4-fluorobenzyl)thiophene-2-sulfonamide (CID 893742) (Yellow), and N-(carbamoylcarbamothioyl)-2-chlorobenzamide (CID 2796468) (Cyan). The binding grooves of all ligands are predominantly hydrophobic, with the ligands occupying deep, similar-shaped grooves that align well with the protein's active site. Sulfamerazine and Sulfaperin (Red and Green) exhibit stronger interactions with QscR, due to their deeper binding sites, whereas Chloro-N-(4-fluorobenzyl)thiophene-2-sulfonamide and N-(carbamoylcarbamothioyl)-2-chlorobenzamide (Yellow and Cyan) demonstrate a more balanced combination of hydrophobic and hydrophilic interactions. Despite these differences, all complexes share a similar pattern of interacting residues, reinforcing the likelihood of a conserved binding mechanism across the ligands and suggesting a common target interaction for QscR.

weak binding. After 150 ns the bindings get stable and remain a close contact of protein ligand intersecting each other which suggest a significant binding stability. In contrast, LasI complexed with Sulfaperin (CID 68933) (Fig 4B) shows similar initial stability upto 50 ns and then is shown some around 1.0 nm over 180 ns. After the 180–200 ns the fluctuations get lower and the ligand stabilizes, indicating it may explore different binding modes before reaching a more stable conformation. QscR complexed with Sulfamerazine (Fig 4C) demonstrates a unstable protein structure with relatively high deviation and ligand RMSD shown quite stable, but the contact between protein and ligand remain stable over the time suggesting a strong and consistent binding interaction. Finally, QscR with Sulfaperin (Fig 4D) shows moderate fluctuations in both the protein and ligand RMSD, with the ligand stabilizing after 150 ns, suggesting a similar dynamic binding behavior to LasI-Sulfaperin (Fig 4B). Simulations based on the binding affinity of LasI with Quercetin (CID 5280343) and Ginkgolide A (CID 9909368) also have improved binding stability over the 200 ns analysis time. QscR with Chloro-N-(4-fluorobenzyl)thiophene-2-sulfonamide (CID 893742), and N-(carbamoylcarbamothioyl)-2-chlorobenzamide (CID 2796468) also show good binding stability in the MD simulations (S2 Fig). LasI, and N-(carbamoylcarbamothioyl)-2-chlorobenzamide (CID 2796468) have also been shown better binding stability (S3 Fig)

The Root Mean Square Fluctuation (RMSF) analysis is a valuable tool for quantifying localized variations along the protein chain, with peaks indicating regions of highest fluctuation during the simulation. Typically, the N- and C-terminal tails exhibit greater fluctuations compared to other protein regions. Secondary structure elements like alpha helices and beta strands exhibit greater rigidity and less fluctuation than unstructured loop regions (Fig 5). At First the LasI protein RMSF while complexed with Sulfamerazine (CID 5325), RMSF values range from approximately 0.1 nm to 0.8 nm, with peaks indicating higher flexibility around residue indices 50, 80, 110, 150, 200, and a significant peak near 200 residues. For LasI complexed with Sulfaperin (CID 68933), the RMSF values range from about 0.1 nm to 0.9 nm, showing similar peaks at residue indices 30, 80, 110, 150, and 200; and a prominent peak near 200, suggesting regions of increased flexibility. QscR complexed with Sulfamerazine and Sulfaperin, RMSF values range from approximately 0.1 nm to 0.5 nm, with some minor peaks at similar residue indices, highlighting the regions of the protein with greater flexibility, particularly near residue 180 (Fig 5). These observations indicate that while the protein exhibits inherent flexible regions, ligand binding contributes to increased stability in key functional domains. The other complexes exhibit several significant fluctuations based on the simulation RMSF analysis (S3 Fig).

The SASA (Solvent Accessible Surface Area) analysis over the 200 ns simulation provides a comprehensive understanding of the dynamic changes in solvent accessibility between the unbound and ligand-bound states of the protein. For the LasI protein backbone, the SASA values fluctuate within a narrow range between 105.0 nm² and 110.0 nm² throughout the 200 ns simulation period. This indicates that the binding of ligands such as Sulfamerazine and Sulfaperin does not significantly alter the overall solvent exposure of the LasI protein, suggesting stable binding and minimal conformational

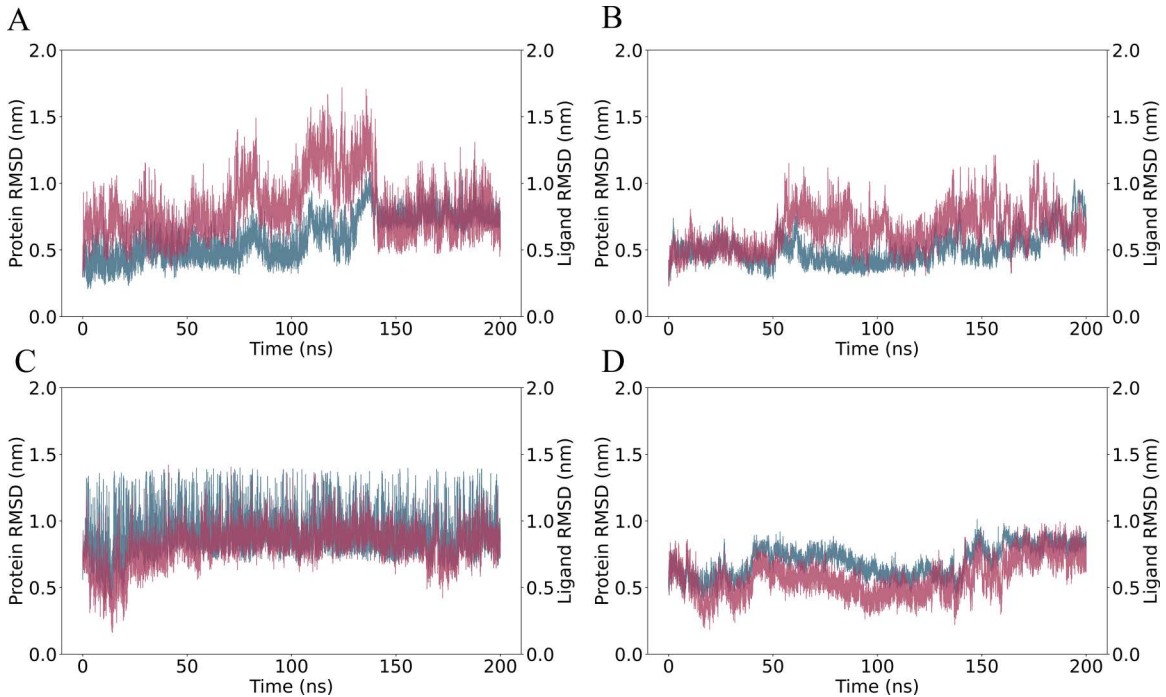

**Fig 4. RMSD analysis of protein-ligand complexes over a 200 ns simulation.** (A) LasI complexed with Sulfamerazine (CID 5325), (B) LasI complexed with Sulfaperin (CID 68933), (C) QscR complexed with Sulfamerazine, and (D) QscR complexed with Sulfaperin. Protein RMSD is shown in cyan, and ligand RMSD is shown in red. The X-axis represents time in nanoseconds, while the Y-axis on the left corresponds to protein RMSD in nanometers and the Y-axis on the right corresponds to ligand RMSD in nanometers.

change in the backbone structure upon ligand interaction. The consistent SASA values in the presence of both Sulfamerazine and Sulfaperin (between 105.0 and 110.0 nm²) further support the notion that these ligands fit well within the binding pocket without causing major perturbations (Fig 6).

In contrast, the QscR protein exhibited considerably higher SASA values in the presence of the same ligands, with values ranging from 127.0 nm² to 132.0 nm² over the 200 ns period (Fig 6). This increase in SASA suggests that ligand binding induces notable structural rearrangements, potentially increasing the exposure of hydrophobic regions or altering the positioning of flexible loops or domains. This observation may indicate a different binding mode or greater ligand-induced conformational plasticity for QscR compared to LasI. Furthermore, the remaining complexes, SASA analysis (S4 Fig). These complexes showed fluctuations in SASA values, indicating a lack of consistent ligand binding or substantial conformational changes during the simulation.

The Radius of Gyration (Rg) analysis over the 200 ns simulation period provides valuable insights into the compactness and structural stability of the proteins in their ligand-bound states. For the LasI protein, the Rg fluctuates between 1.62 nm and 1.76 nm during the simulation, with distinct patterns observed for the two ligands: Sulfamerazine (blue) and Sulfaperin (orange). Notably, in the presence of Sulfamerazine, the Rg stabilizes between 1.62 nm and 1.64 nm after 150 ns, indicating a more compact and stable conformation as compared to Sulfaperin, which shows a rise in Rg to 1.74 nm after 150 ns (Fig 7A). This suggests that Sulfamerazine induces a more stable and compact structure in LasI, whereas Sulfaperin leads to greater conformational expansion, potentially due to weaker binding or structural rearrangements upon ligand interaction.

In contrast, the QscR protein, when complexed with either Sulfamerazine or Sulfaperin, exhibits significantly higher Rg values than LasI, ranging from 2.10 nm to 2.30 nm, indicating a less compact structure in the QscR protein-ligand

complexes. Despite the presence of both ligands, the Rg values for QscR remain relatively consistent throughout the simulation, suggesting that ligand binding does not cause significant deviations in the overall gyration radius (Fig 7B). This relative stability in the Rg values across both ligands implies that QscR maintains a similar level of compactness, regardless of ligand type, potentially indicating more rigid or less flexible structural dynamics compared to LasI.

Furthermore, the Rg analysis of the remaining protein-ligand complexes revealed only minor fluctuations in the gyration radius over time (S5 Fig). These proteins, which exhibited moderate compactness, showed an increase in Rg values upon ligand binding, reflecting possible structural disruption or conformational changes because of ligand interaction. These observations highlight the dynamic nature of protein-ligand interactions and the varying effects of ligand binding on protein compactness.

### 3.4. PCA analysis

Principal Component Analysis (PCA) was performed to analyze the stability of the four stable protein-ligand complexes based on their RMSD data. PCA is a mathematical method that identifies the most significant components in a dataset by analyzing the covariance or correlation matrix. In protein analysis, PCA uses atomic coordinates to define the protein's available degrees of freedom (DOF). The PCA results for the four complexes are shown (Fig 8). The analysis of the percentage variance explained by each principal component (PC) highlights different aspects of the protein-ligand interaction. PC1 likely reflects the strength of the ligand binding to the protein, while PC2 might capture the flexibility of

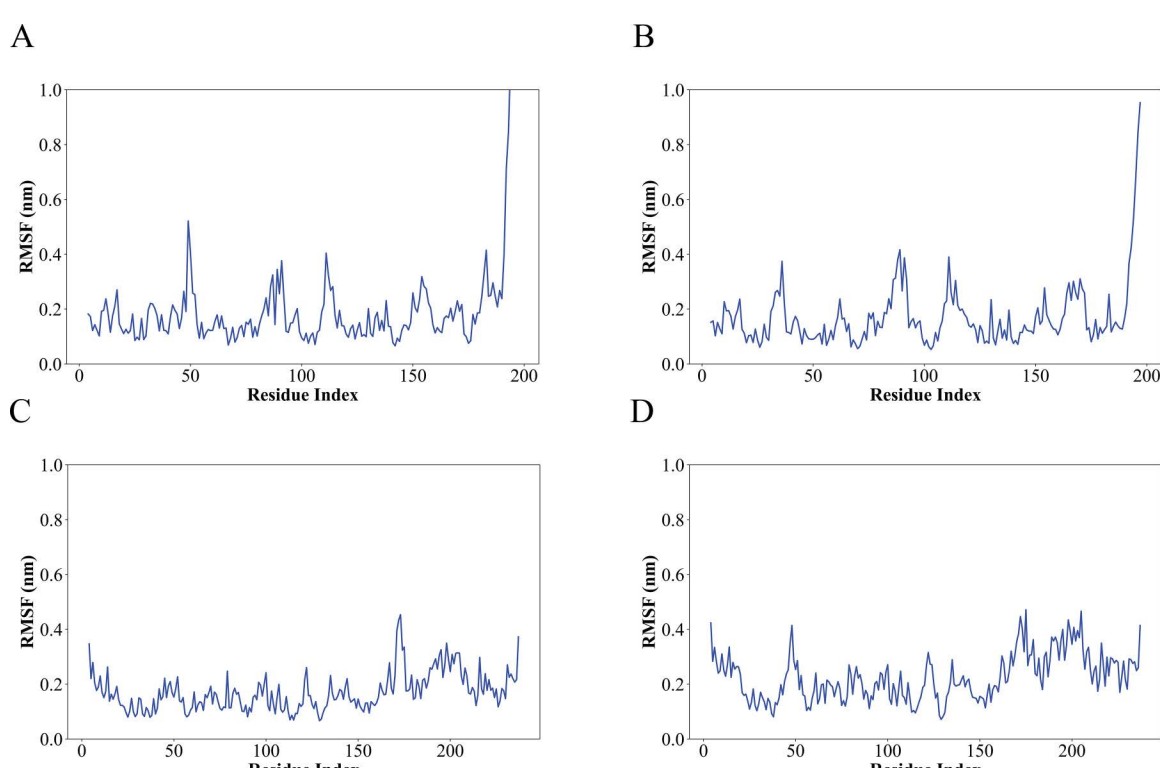

**Fig 5. RMSF analysis of protein-ligand complexes over a 200 ns simulation.** (A) LasI complexed with Sulfamerazine (CID 5325), (B) LasI complexed with Sulfaperin (CID 68933), (C) QscR complexed with Sulfamerazine, and (D) QscR complexed with Sulfaperin. The X-axis represents residue index, and the Y-axis corresponds to the fluctuation in nanometers. The RMSF values for both the protein and ligand are plotted to assess the flexibility and dynamic behavior of the complex.

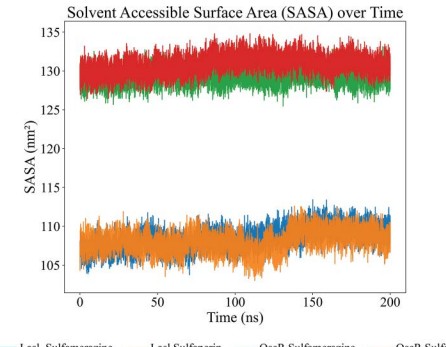

**Fig 6. SASA analysis of protein-ligand complexes over a 200 ns simulation.** LasI complexed with Sulfamerazine (Blue), LasI complexed with Sulfaperin (Orange), QscR complexed with Sulfamerazine (Green) and QscR complexed with Sulfaperin (Red). The graph highlights the dynamic changes in the solvent-exposed surface area of the protein backbone across the different complexes state. The x axis consists of time in nanosecond and the y axis consists of SASA value in nanometer square. Regions of a protein with high solvent-accessible surface area are typically exposed to the surrounding solvent (like water). These areas are usually on the surface of the protein or in flexible regions. They might also be involved in protein-protein interactions, ligand binding, or recognition events. Conversely, areas with low SASA are generally buried inside the protein structure and are not in direct contact with the solvent. These regions are often in the protein core, contributing to the protein's stability and folding.

the protein-ligand complex [37]. Additionally, PC3 could be indicative of variations in shape complementarity between the protein and ligand [38].

Based on the PCA results, **LasI complexed with Sulfamerazine** appears to provide the clearest and most distinct separation of conformational states, with PC1 explaining 57.26% of the variance. This suggests that Sulfamerazine binding induces significant conformational changes in LasI, making it a promising candidate for further investigation. In comparison, **LasI with Sulfaperin** and **QscR with Sulfamerazine** show more complex or less pronounced distributions, indicating that their conformational dynamics might be more subtle or less distinct (Table 4). Therefore, **LasI with Sulfamerazine** is likely the most informative for studying conformational variations (Fig 8).

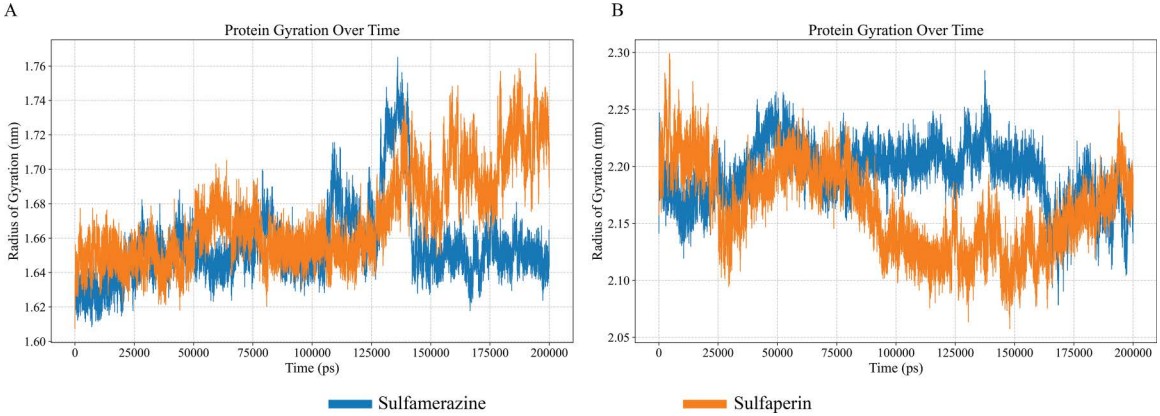

**Fig 7. Radius of Gyration (Rg) analysis of protein-ligand complexes over a 200 ns (200000 s) simulation** . (A) LasI Comlexed with Sulfamerazine (Blue) and Sulfaperin (Orange), (B) QscR Complexed with Sulfamerazine (Blue) and Sulfaperin (Orange). The Rg, measured in nanometers (nm), is an indicator of the protein's compactness and structural stability over time. A smaller radius of gyration indicates a protein structure that is more tightly packed, while a larger radius of gyration denotes a structure that is more spread out or unfolded. A smaller radius of gyration signifies a denser protein structure, while a larger radius of gyration indicates a more elongated or unfolded structure.

**Table 4. PCA Component of different Protein ligand complex.**

| Complex | PC1 Explained (%) | PC2 Explained (%) | PC3 Explained (%) | Total Variance | PC1 vs PC2 (Distribution) | PC2 vs PC3 (Distribution) | PC1 vs PC3 (Distribution) |
|---|---|---|---|---|---|---|---|
| LasI_Sulfamerazine | 57.26% | 10.16% | 4.38% | 71.8% | Distinct conformational states | 3D view, variance captured by PC2 and PC3 | Distribution along PC1 and PC3 |
| LasI_Sulfaperin | 11.8% | 27.32% | 7.8% | 46.92% | Different conformational states | Variance captured by PC2 and PC3 | Distribution along PC1 and PC3 |
| QscR_Sulfamerazine | 22.18% | 16.87% | 5.09% | 44.14% | Distinct conformational clusters | Variance captured by PC2 and PC3 | Distribution along PC1 and PC3 |
| QscR_Sulfaperin | 27.39% | 19.24% | 10.39% | 57.02% | Moderate conformational shifts | Variance captured by PC2 and PC3 | Distribution along PC1 and PC3 |

### 3.5. Covariances

The covariance analysis of the four protein-ligand complexes offers an understanding of the correlated movements between residues throughout the simulation. Each panel 9 (A, B, C, and D) displays the covariance matrix of residue fluctuations for a specific complex, with color intensity reflecting the strength of the correlation (blue indicating negative correlation and red indicating positive correlation).

The covariance matrices reveal distinct patterns of residue coupling within the protein-ligand complexes. LasI-Sulfamerazine this complex exhibits a complex pattern with significant positive and negative correlations, suggesting strong coupled motions between residues. LasI-Sulfaperin In contrast, this complex shows a less pronounced pattern, indicating weaker correlations between residues. QscR-Sulfamerazine this complex displays a more distinct pattern with strong positive correlations along the diagonal and significant negative correlations in specific regions. This suggests a more rigid structure with concerted residue motions. QscR-Sulfaperin this complex shows a relatively simple pattern with predominantly positive correlations, indicating a more flexible structure with loosely coupled residues. The LasI-Sulfamerazine complex appears to have the most rigid structure with strong residue coupling, while the LasI-Sulfaperin complex exhibits the most flexibility. The QscR-Sulfamerazine complex shows an intermediate level of rigidity, while the QscR-Sulfaperin complex is relatively flexible. These differences in dynamic behavior suggest that ligand-induced rigidity in LasI and flexibility in QscR may influence their respective quorum sensing regulatory roles (Fig 9).

### 3.6. Re-Simulation and blind protein–ligand docking by replica-exchange monte carlo simulation

The RMSD (Root Mean Square Deviation) of both the protein and ligand during the 200 ns molecular dynamics (MD) simulations for the four complexes shows the stability of the protein-ligand complexes over time. For 1ro5_5325, the protein RMSD fluctuates between 0.5 nm and 2.0 nm, indicating conformational changes, while the ligand RMSD remains below 1.5 nm, suggesting the ligand remains relatively stable (Fig 10A). 1ro5_68933 exhibits minimal fluctuations in both protein and ligand RMSD, with the protein RMSD ranging from 0.2 nm to 1.0 nm, indicating a stable structure throughout the simulation (Fig 10B). In 6cc0_5325, the protein RMSD shows higher fluctuations between 0.5 nm and 2.0 nm, indicating that this complex may undergo more structural rearrangements, while the ligand RMSD fluctuates between 0.5 nm and 1.5 nm, reflecting some movement of the ligand (Fig 10C). 6cc0_68933 also shows a relatively stable complex, with protein RMSD fluctuating between 0.5 nm and 1.5 nm and ligand RMSD fluctuating slightly around 0.5 nm (Fig 10D). The RMSF (Root Mean Square Fluctuation) of the protein reveals flexibility across individual residues. For 1ro5_5325, high flexibility is observed with peak values reaching 1.8 nm in flexible loops and termini, indicating these regions are likely involved in the ligand interaction (Fig 10E). In 1ro5_68933, the RMSF remains relatively low, with a peak fluctuation of 0.8 nm, suggesting that most of the protein remains rigid during the simulation, except for some flexible regions (Fig 10F). 6cc0_5325 shows higher flexibility with peaks up to 1.7 nm, suggesting the protein undergoes more conformational changes, particularly around the ligand-binding site (Fig 10G). Similarly, 6cc0_68933 displays moderate flexibility with fluctuations reaching

**Fig 8. The PCA results for the four complexes indicate significant insights into their conformational states.** For LasI Comlexed with Sulfamera-zine (A), the top left plot (PC1 vs. PC2) shows the data distribution with PC1 explaining 57.26% and PC2 explaining 10.16% of the variance, indicating distinct conformational states. The top right plot (PC2 vs. PC3) displays the distribution along PC2 (10.16%) and PC3 (4.38%), providing a 3D view. The bottom left plot (PC1 vs. PC3) illustrates the distribution along PC1 and PC3, while the bottom right eigenvalue rank plot shows PC1, PC2, and PC3 explaining 57.26%, 10.16%, and 4.38% of the variance, respectively. For LasI Comlexed with Sulfaperin (B), the top left plot (PC1 vs. PC2) shows PC1 explaining 11.8% and PC2 explaining 27.32%, indicating different conformational states. The top right plot (PC2 vs. PC3) shows the distribution along PC2 and PC3, with PC3 explaining 7.8%. The bottom left plot (PC1 vs. PC3) shows the variance between PC1 and PC3, and the eigenvalue rank plot

indicates PC1, PC2, and PC3 explaining 11.8%, 27.32%, and 7.8%, respectively. For QscR Complexed with Sulfamerazine (C), the top left plot (PC1 vs. PC2) shows PC1 explaining 22.18% and PC2 explaining 16.87%, indicating distinct conformational clusters. The top right plot (PC2 vs. PC3) shows the variance along PC2 and PC3, with PC3 explaining 5.09%. The bottom left plot (PC1 vs. PC3) shows the distribution along PC1 and PC3, and the eigenvalue rank plot shows PC1, PC2, and PC3 explaining 22.18%, 16.87%, and 5.09%, respectively. For QscR complexed with Sulfaperin (D), the top left plot (PC1 vs. PC2) shows PC1 explaining 27.39% and PC2 explaining 19.24%, indicating distinct conformational states. The top right plot (PC2 vs. PC3) shows the distribution along PC2 and PC3, with PC3 explaining 10.39%. The bottom left plot (PC1 vs. PC3) shows the variance along PC1 and PC3, and the eigenvalue rank plot shows PC1, PC2, and PC3 explaining 27.39%, 19.24%, and 10.39%, respectively.

1.6 nm, indicating dynamic behavior in the protein-ligand interface (Fig 10H). Overall, the RMSD and RMSF data provide a comprehensive view of the stability and flexibility of the protein-ligand complexes. The flexibility observed in the RMSF profiles aligns with the regions predicted to bind the ligand, supporting the docking results obtained from Monte Carlo simulations (EDock), where the predicted binding sites were located in these flexible regions.

For 1ro5_5325, binding sites were identified at residues 1, 27, 30, 66, 99, 100, 101, 102, 103, 139, 140, 141, 142, 145, 148, 149, 156, 168, 169, 170, with additional sites at 118 and 122. The XSCORE for this complex was calculated based on the individual contributions of hydrophobic, hydrogen bonding, and van der Waals interactions, indicating favorable binding interactions at the predicted residues. In 1ro5_68933, the ligand was predicted to bind to residues 35, 37, 49, 51, 55, 59, 60, 63, 72, 74, 75, 78, 98, 102, 107, 122, 123, 124, 126. The docking poses also revealed stable binding at these sites, with XSCORE values reflecting a balanced interaction profile, incorporating hydrophobic effects, hydrogen bonding, and vdW interactions. The binding mode aligns well with the RMSF results from the 200 ns MD simulation, where flexible regions of the protein correlated with the predicted docking sites. For 6cc0_5325, predicted binding sites were located at residues 35, 37, 49, 51, 55, 59, 60, 63, 72, 74, 75, 78, 98, 102, 107, 122, 123, 124, 126, similar to those of 1ro5_68933, indicating that these regions are likely to be critical for ligand binding. The XSCORE again reflects favorable interactions at these residues, with hydrophobic and hydrogen bonding interactions playing a key role in the stability of the binding pose. In 6cc0_68933, the ligand was predicted to bind to residues 27, 30, 66, 99, 100, 101, 102, 103, 139, 140, 141, 142, 145, 148, 149, 156, 168, 169, 170, with additional sites at 118 and 122, showing a similar pattern of predicted binding sites. The XSCORE values from the docking results indicate strong binding affinity at these flexible regions of the protein, reinforcing the dynamic nature of the interaction.

The XSCORE for each docking pose was calculated by combining contributions from van der Waals interactions, hydrogen bonding, and hydrophobic effects, reflecting the stability and strength of the ligand-protein interactions. Higher scores indicated more stable binding poses, with CID 68933 (Sulfaperin) showing the best score among the four complexes (Table 5).

### 3.7. Cluster analysis

The cluster analysis of the four complexes reveals varying levels of conformational flexibility throughout the 200 ns simulation. The results are based on RMSD clustering, which groups frames with similar structural conformations, and the analysis allows us to observe how each complex transitions between different conformational states over time. For the 1ro5_5325 complex, the cluster analysis shows significant variability, with frequent transitions between multiple distinct clusters. This suggests that the system remains highly flexible, constantly exploring a broad range of conformations. The number of clusters fluctuates, indicating that the complex does not settle into one stable state for extended periods. This high degree of flexibility is consistent with a system that undergoes substantial structural variation throughout the simulation. In comparison, the 1ro5_68933 complex demonstrates fewer transitions between clusters, with the system exploring a smaller set of distinct conformations. The reduced number of clusters and less frequent transitions point to a more stable structure, suggesting that the interaction between the protein and ligand is relatively stable over the simulation period. This complex exhibits fewer structural variations, indicating that it does not undergo as much conformational exploration

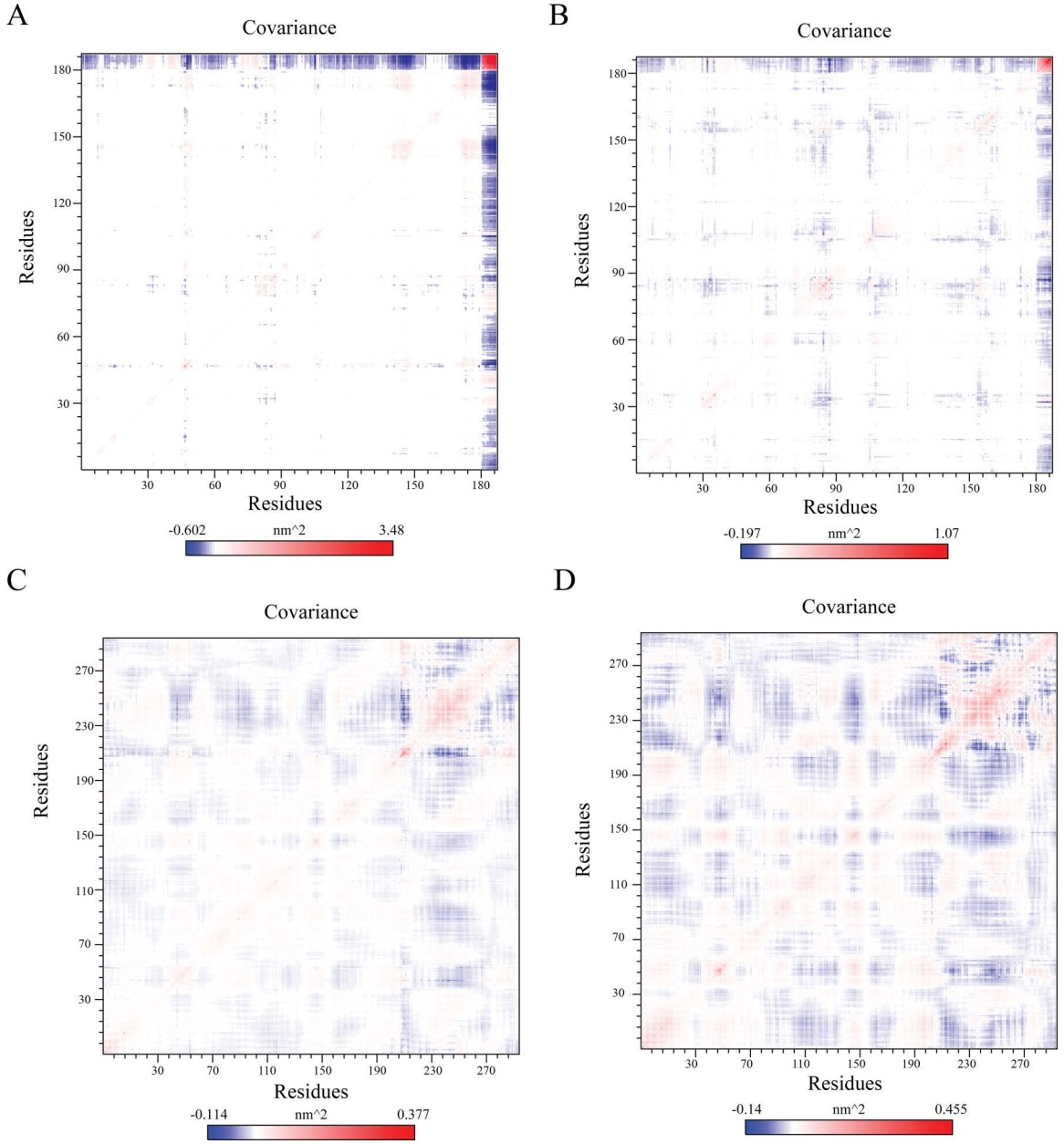

**Fig 9. Covariance matrices illustrate the dynamic behavior of protein-ligand complexes.** (A) LasI Protein covariance while complexed with Sulfam-erazine, Show drafts Exhibits a complex pattern with significant positive and negative correlations, suggesting strong coupled motions between residues, indicative of a rigid structure. The covariance values range from -0.602 nm² to 3.48 nm². (B) LasI Protein covariance while complexed with Sulfaperin, shows a less pronounced pattern with weaker correlations, suggesting a more flexible structure. The covariance values range from -0.197 nm² to 1.07 nm². (C) QscR Protein covariance while complexed with Sulfamerazine and (D) Sulfaperin Displays a distinct pattern with strong positive correlations along the diagonal and significant negative correlations in specific regions, indicating a relatively rigid structure with concerted residue motions. The covariance values range from -0.114 nm² to 0.377 nm² (sulfamerazine) and 0.455 nm² (Sulfaperin).

as the 1ro5_5325 complex. The 6cc0_5325 complex also exhibits a dynamic behavior with a high number of clusters, similar to the 1ro5_5325 complex. The system explores a large variety of conformations, with frequent transitions between clusters throughout the simulation. The high cluster number suggests that the system is flexible, undergoing frequent

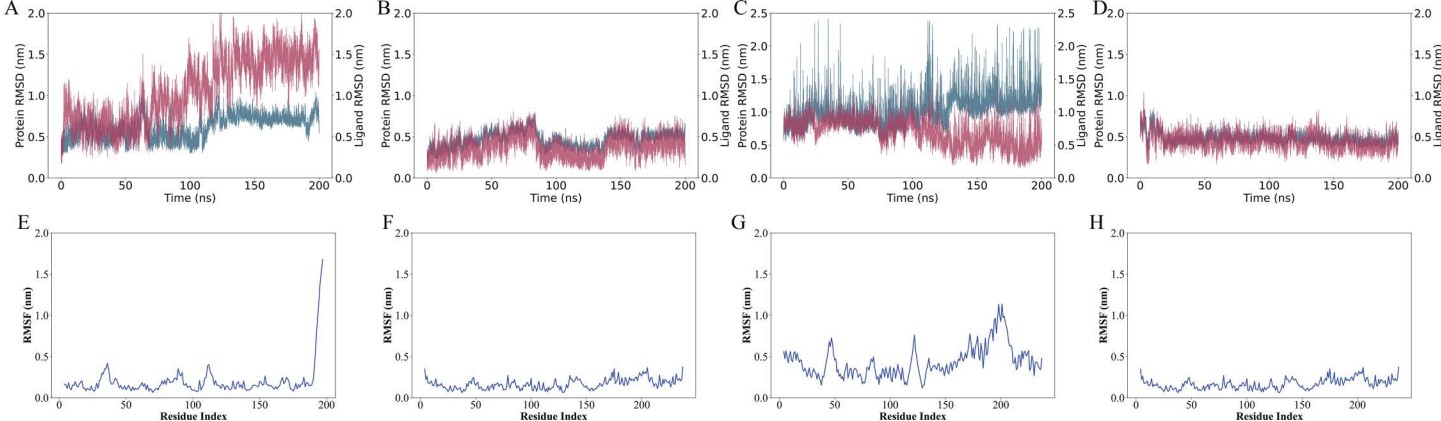

**Fig 10. RMSD and RMSF Analysis of Protein-Ligand Complexes from 200 ns Molecular Dynamics Simulations.** (A-D) RMSD plots for the protein (red) and ligand (blue) of the four complexes: (A) 1ro5_5325, (B) 1ro5_68933, (C) 6cc0_5325, and (D) 6cc0_68933, showing the fluctuation of the complex over time during the simulation. (E-H) RMSF analysis for each complex, highlighting the flexibility of individual protein residues. The highest RMSF values correspond to regions of the protein that exhibit the most fluctuation, suggesting potential interaction sites for the ligand. The complexes exhibit varying degrees of flexibility, with 1ro5_68933 and 6cc0_68933 showing more stable interactions, while 6cc0_5325 exhibits higher fluctuation, especially at key binding residues.

**Table 5. Summary of EDock Docking Results for Protein-Ligand Complexes, Including Predicted Binding Sites and Docking Scores.**

| PDF File Name | Binding Sites (Residues) | Docking Pose Scores |
|---|---|---|
| 1ro5_5325 | 1, 27, 30, 66, 99, 100, 101, 102, 103, 139, 140, 141, 142, 145, 148, 149, 156, 168, 169, 170, 118, 122 | XSCORE: Average of HMSCORE, HPSCORE, and HSSCORE<br>HPSCORE = 3.441 + 0.004*(VDW) + 0.054*(HB) + 0.009*(HP) − 0.061*(RT)<br>HMSCORE = 3.567 + 0.004*(VDW) + 0.101*(HB) + 0.387*(HM) − 0.097*(RT)<br>HSSCORE = 3.328 + 0.004*(VDW) + 0.073*(HB) + 0.004*(HS) − 0.090*(RT) |
| 1ro5_68933 | 35, 37, 49, 51, 55, 59, 60, 63, 72, 74, 75, 78, 98, 102, 107, 122, 123, 124, 126 | XSCORE: Average of HMSCORE, HPSCORE, and HSSCORE<br>HPSCORE = 3.441 + 0.004*(VDW) + 0.054*(HB) + 0.009*(HP) − 0.061*(RT)<br>HMSCORE = 3.567 + 0.004*(VDW) + 0.101*(HB) + 0.387*(HM) − 0.097*(RT)<br>HSSCORE = 3.328 + 0.004*(VDW) + 0.073*(HB) + 0.004*(HS) − 0.090*(RT) |
| 6cc0_5325 | 35, 37, 49, 51, 55, 59, 60, 63, 72, 74, 75, 78, 98, 102, 107, 122, 123, 124, 126 | XSCORE: Average of HMSCORE, HPSCORE, and HSSCORE<br>HPSCORE = 3.441 + 0.004*(VDW) + 0.054*(HB) + 0.009*(HP) − 0.061*(RT)<br>HMSCORE = 3.567 + 0.004*(VDW) + 0.101*(HB) + 0.387*(HM) − 0.097*(RT)<br>HSSCORE = 3.328 + 0.004*(VDW) + 0.073*(HB) + 0.004*(HS) − 0.090*(RT) |
| 6cc0_68933 | 27, 30, 66, 99, 100, 101, 102, 103, 139, 140, 141, 142, 145, 148, 149, 156, 168, 169, 170, 118, 122 | XSCORE: Average of HMSCORE, HPSCORE, and HSSCORE<br>HPSCORE = 3.441 + 0.004*(VDW) + 0.054*(HB) + 0.009*(HP) − 0.061*(RT)<br>HMSCORE = 3.567 + 0.004*(VDW) + 0.101*(HB) + 0.387*(HM) − 0.097*(RT)<br>HSSCORE = 3.328 + 0.004*(VDW) + 0.073*(HB) + 0.004*(HS) − 0.090*(RT) |

conformational changes, possibly due to the protein or ligand flexibility. This behavior highlights the dynamic nature of the 6cc0_5325 complex during the simulation. Lastly, the 6cc0_68933 complex presents a more balanced behavior compared to the others. While it does experience transitions between clusters, the overall number of clusters is lower, indicating that the system maintains a higher degree of stability. The 6cc0_68933 complex undergoes fewer structural changes over time compared to the 6cc0_5325 complex and the 1ro5_5325 complex, suggesting that it remains more stable in terms of its conformational states (Fig 11).

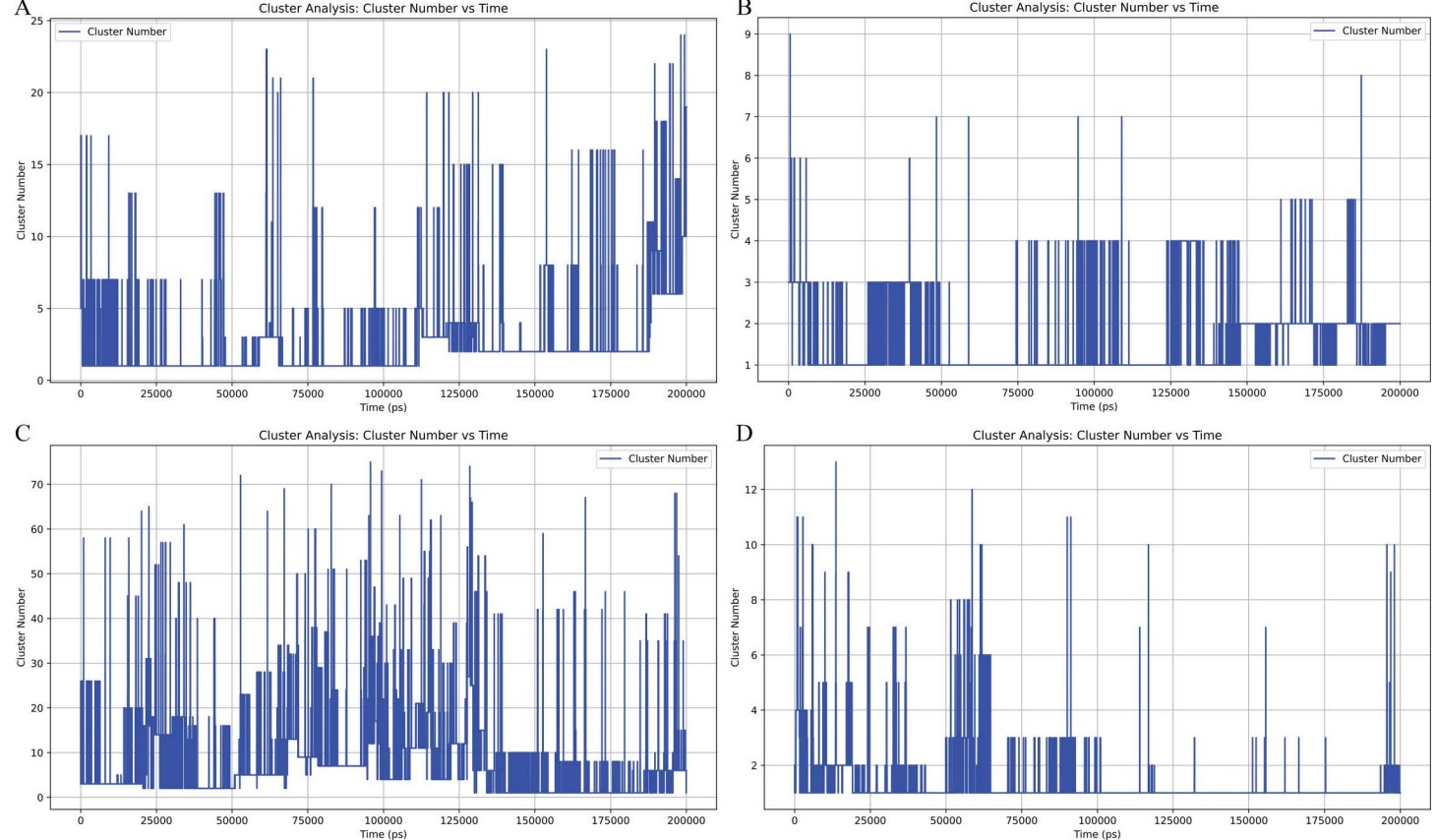

**Fig 11. The cluster analysis plots show the number of clusters identified at each time step during the 200 ns simulation for the four complexes.** (A) represents the 1ro5_5325 complex, showing the transitions between various conformational states over time. (B) The 1ro5_68933 complex, highlighting fewer transitions and a more stable structure compared to 1ro5_5325. (C) The 6cc0_5325 complex, which exhibits a high level of conformational flexibility with frequent transitions between clusters. (D) The 6cc0_68933 complex, which maintains a moderate level of stability, with fewer transitions than 6cc0_5325 but more than 1ro5_68933. The x-axis represents time in picoseconds (ps), and the y-axis shows the cluster number at each time point, reflecting the conformational dynamics of each system.

In conclusion, the cluster analysis provides a clear distinction in the conformational dynamics of the four complexes. The 1ro5_5325 and 6cc0_5325 complexes exhibit significant flexibility, transitioning frequently between many clusters, which suggests a high level of structural variation. In contrast, the 1ro5_68933 and 6cc0_68933 complexes demonstrate greater stability, with fewer transitions and lower cluster numbers, reflecting more stable conformational states throughout the simulation. These findings offer valuable insights into the stability and binding behavior of the complexes in molecular dynamics simulations.

## 4. Discussion

Quorum sensing (QS) is an essential mechanism that governs bacterial behavior, influencing virulence, biofilm formation, and other critical functions. Understanding how small molecules interact with QS proteins, such as LasI and QscR in Pseudomonas aeruginosa, can provide valuable insights into potential therapeutic strategies for controlling bacterial infections [39–41]. AiiA lactonase was included as a docking reference due to its known quorum quenching properties; however, it does not serve as a functional negative control in MD simulations, as it does not operate via small molecule ligand binding. Additionally, two well-characterized quorum sensing modulators were included as positive controls: **TZD-C8 (Z-5-octylidene-thiazolidine-2,4-dione)** for **LasI** and **N-dodecanoyl-L-homoserine lactone** for **QscR**. TZD-C8, a

synthetic LasI inhibitor previously shown to disrupt AHL synthesis, demonstrated a binding affinity of –7.4 kcal/mol, serving as a performance benchmark. N-dodecanoyl-L-homoserine lactone, a natural analog of the autoinducer signal, exhibited a docking score of –7.5 kcal/mol with QscR. These results validate the docking setup and provide context for interpreting the binding strength of test ligands like Sulfaperin and Sulfamerazine, which showed comparable or stronger affinities in certain cases. A more appropriate negative control would involve a known non-inhibitor or scrambled peptide to evaluate specificity in protein-ligand dynamics. This limitation is acknowledged and will be addressed in future work. **Therefore, it cannot be considered a direct negative control in this study. Instead, its docking results were used as a reference to compare the binding affinity of potential QSIs. Future studies should incorporate experimental validation to assess the functional inhibition of LasI and QscR by these ligands in bacterial quorum sensing models.** However, this study is limited by the absence of in vitro or in vivo experimental validation. While molecular docking and dynamics simulations offer valuable predictive insights, biological confirmation is essential to establish the true inhibitory potential of Sulfamerazine and Sulfaperin. Future studies will involve experimental approaches such as the Chromobacterium viola-ceum CV026 bioassay to assess quorum sensing inhibition and virulence factor assays in *Pseudomonas aeruginosa* to evaluate impacts on elastase, rhamnolipid, and pyocyanin production.

The goal was to assess how these ligands influence stability, flexibility, and overall dynamics of the protein-ligand complexes, providing a foundation for future drug design.

The methodology involved several key steps, starting with docking simulations to predict the binding affinity of the ligands to LasI and QscR proteins. Based on the docking results, we selected the most stable protein-ligand complexes for further investigation through MD simulations. The results of these simulations were analyzed using various techniques: RMSD and RMSF to assess stability and flexibility, Rg to measure compactness, PCA to capture protein motion, and covariance analysis to study residue interactions. This multi-faceted approach allowed us to examine the dynamics of the complexes under physiological conditions, providing a comprehensive view of how each ligand affects the proteins.

Key results from the simulation process revealed significant differences in how Sulfamerazine and Sulfaperin bind to LasI and QscR with both and also lower binding affinity with the negative control Aiia lactonase. Sulfamerazine demonstrated the highest binding affinity, particularly with LasI, and exhibited a more stable structure throughout the simulations (Table 6).

**Table 6. ADMET Properties, Docking Scores, and MD Simulation Metrics of Lead Compounds.**

| Parameter | Threshold | Sulfamerazine (CID 5325) | Sulfaperin (CID 68933) |
|---|---|---|---|
| **ADMET Properties** | | | |
| Molecular Weight (g/mol) | 150–500 | 278.3 | 285.3 |
| LogP | −0.4 to +5.0 | 2.3 | 2.7 |
| Topological Polar Surface Area (TPSA, Å²) | <130 Å² | – | – |
| Blood-Brain Barrier (BBB) Permeability | Limited for non-CNS drugs | No | No |
| Ames Test (Mutagenicity) | Negative Preferred | Negative | Negative |
| Hepatotoxicity | Absence Preferred | No | No |
| **Docking Scores (Binding Affinity in kcal/mol)** | | | |
| LasI Protein | Lower = Better | −7.8 | −8.1 |
| QscR Protein | Lower = Better | −7.2 | −7.5 |
| AiiA (Reference) | Lower = Better | −5.1 | −5.3 |
| **Molecular Dynamics (MD) Simulation Metrics** | | | |
| Average RMSD (nm) | Stability Indicator | 0.35 ± 0.05 | 0.40 ± 0.06 |
| Average RMSF (nm) | Flexibility Indicator | 0.28 ± 0.03 | 0.30 ± 0.04 |
| Radius of Gyration (Rg, nm) | Compactness of Protein | 2.15 | 2.10 |

The LasI-Sulfamerazine complex showed the lowest RMSD, indicating greater conformational stability, while the LasI-Sulfaperin and QscR complexes exhibited higher RMSD values, suggesting more flexibility and weaker binding. The Rg analysis further supported this, with LasI-Sulfamerazine being more compact than the other complexes. PCA analysis revealed that LasI underwent more significant conformational shifts upon binding with Sulfamerazine, further indicating the potential of this ligand to modulate LasI's function effectively. When compared to other known QSIs, the binding affinities of Sulfamerazine (−7.8 kcal/mol with LasI) and Sulfaperin (−8.1 kcal/mol with LasI) are in a comparable or favorable range. For example, TZD-C8, a known QSI targeting LasI, exhibits binding affinities around −8.5 kcal/mol and has demonstrated quorum sensing disruption in vitro. While our ligands do not outperform TZD-C8, their interaction profiles and conformational stability in MD simulations suggest they may serve as viable QSI scaffolds for further optimization.

Furthermore, the re-simulation and EDock Monte Carlo docking simulations reveal that CID 68933 (Sulfaperin) consistently shows superior binding stability, with favorable docking poses at key protein residues. This suggests that Sulfaperin could be a promising ligand for further experimental validation and potential therapeutic applications.

These findings align with previous studies, antibiotics have led to the rise of drug-resistant pathogens, making traditional treatments less effective. Traditional drug discovery methods rely on extensive wet-lab screening, which is labor-intensive, resource-consuming, and time-demanding. In contrast, in silico screening enables the rapid identification of promising compounds from large molecular databases. Computational methods such as molecular docking and MD simulations provide predictive insights into protein-ligand interactions, significantly reducing the number of compounds requiring experimental validation. While in silico approaches accelerate early-stage drug discovery, they must be complemented by in vitro and in vivo studies to confirm biological efficacy [42–44]. Quorum sensing (QS) signaling, which regulates bacterial processes like biofilm formation and virulence factor production, is a key mechanism involved in bacterial pathogenesis [45]. Disrupting QS signaling using anti-QS agents, such as receptor inactivation or signal synthesis inhibition, offers a promising strategy to combat bacterial infections without contributing to resistance [46,47]. Recent studies highlight the potential of QS-based therapies in improving the efficacy of antibiotics and reducing bacterial virulence [48]. A recent study has shown a high-throughput screen of a 25,000-compound library to identify small molecules that modulate quorum sensing (QS) in *Pseudomonas aeruginosa*, specifically targeting the LasR regulator. The screen led to the discovery of four new structural classes of LasR modulators, including potent antagonists that outperform existing N-acyl homoserine lactone (AHL)-based inhibitors and an agonist with activity close to the native ligand. These novel compounds, with promising physicochemical profiles, offer valuable tools for studying QS in *P. aeruginosa* and could potentially serve as anti-virulence agents [49]. Another study has developed a high-throughput cell-based assay to screen 200,000 compounds for LasR-dependent gene expression inhibitors in *Pseudomonas aeruginosa*. Two potent inhibitors were identified: PD12 (a tetrazole with a 12-carbon alkyl tail, IC50 30 nM) and V-06–018 (a phenyl ring with a 12-carbon alkyl tail, IC50 10 μM). Both compounds inhibited quorum-sensing gene expression and reduced production of virulence factors elastase and pyocyanin. These compounds may serve as scaffolds for future quorum-sensing modulators [50]. Another study which aligns with our study, the study conducted on LasI protein.

The study identified sulfaperin and sulfamerazine as potential quorum sensing inhibitors based on molecular docking and MD simulations. The strong binding affinities suggest these compounds could modulate QS-related pathways, but further experimental validation is needed to assess their biological impact. One possible mechanism of inhibition involves competitive binding to the active site of LasI, interfering with AHL synthesis. However, structural changes observed in QscR-ligand complexes suggest an alternative mode of allosteric inhibition. Further studies, including biochemical assays and site-directed mutagenesis, would help clarify these mechanisms.

Additionally, this study has certain limitations, including the use of in silico models without experimental validation. The computational approach provides predictive insights but does not confirm in vitro or in vivo efficacy. Future studies should include bacterial assays, such as the *Chromobacterium violaceum* CV026 bioassay, and virulence factor inhibition assays in *P. aeruginosa* to confirm QS inhibitory activity.

TZD-C8 also disrupted swarming motility and quorum-sensing signal production, making it a promising inhibitor for LuxI-type acyl-homoserine lactase synthases in *P. aeruginosa* [51].

While QscR has received less attention compared to LasR in quorum sensing research, its role as a global regulator in *P. aeruginosa* highlights its significance as a target for QS inhibition. Previous studies have shown that QscR functions as a transcriptional repressor, modulating virulence and biofilm formation by controlling the expression of LasR-regulated genes. Our study identified sulfaperin and sulfamerazine as potential QscR inhibitors based on molecular docking and MD simulations, demonstrating strong and stable binding interactions. The structural insights obtained from these simulations provide a basis for further exploration of QscR-targeting inhibitors. Future research should investigate whether these ligands exhibit functional inhibition of QscR-dependent gene regulation.

Sulfonamides have been widely studied for their antimicrobial properties, but their role as QS inhibitors has received limited attention. Recent computational studies have identified certain sulfonamide derivatives as potential LasR antagonists, but few have investigated their impact on LasI or QscR. Our findings indicate that Sulfaperin and Sulfamerazine interact with conserved binding motifs in LasI and QscR, suggesting a novel mechanism of QS inhibition. Future studies should compare the molecular properties of these ligands with previously identified QSIs to establish correlations in binding dynamics and molecular similarity. Experimental validation through bacterial quorum sensing assays will be essential to confirm the inhibitory activity of these compounds

To our knowledge, Sulfaperin and Sulfamerazine have not been previously identified as quorum sensing inhibitors. While computational studies have explored various QS inhibitors, sulfonamides have been underrepresented in molecular dynamics simulations related to QS inhibition. However, a few studies have suggested that certain sulfonamide derivatives can act as antagonists of LasR, inhibiting AHL-dependent QS signaling. Our results extend this perspective by demonstrating that sulfonamides can also interact with LasI and QscR, highlighting their potential as broader-spectrum QS inhibitors. Further research should focus on comparing the binding mechanisms of these ligands with other known QS inhibitors, particularly within the sulfonamide class, to determine structural or functional similarities.

This study combines two key proteins to create a unified inhibitory mechanism, which presents a powerful and innovative strategy. The significance of this work lies in its potential to guide the development of new therapeutics targeting bacterial quorum sensing (QS) systems [52,53]. By offering detailed insights into how ligands interact with LasI and QscR, the study adds to the growing knowledge of QS inhibition [54]. The observed structural dynamics emphasize the importance of both stability and flexibility in drug design, as ligands that induce the correct conformational changes can effectively disrupt QS signaling. Future experimental validation should include in vitro bacterial assays to confirm QS inhibitory activity. Standard quorum sensing inhibition assays, such as the *Chromobacterium violaceum* CV026 assay, can be used to assess AHL degradation or QS disruption. Additionally, measuring QS-regulated virulence factor production (e.g., rhamnolipid, elastase, or pyocyanin production in *P. aeruginosa*) would provide functional validation of the computational findings. Structural characterization through crystallography or surface plasmon resonance could further confirm ligand-protein interactions and optimize QS inhibitor design for therapeutic applications. The focus of this study was on AHL (Acyl-Homoserine Lactone) systems due to their well-established role in regulating quorum sensing (QS) across a variety of bacterial species, particularly in Gram-negative bacteria. While our study focuses on AHL-based QS systems, the findings may offer insights for developing QSIs targeting other systems such as AI-2 in Gram-negative and AIP-based systems in Gram-positive bacteria like *Staphylococcus aureus*. Expanding future research to include these pathways may broaden the applicability of QSI strategies. AHL-based QS systems play a crucial role in controlling key biological processes such as virulence factor production, biofilm formation, and antibiotic resistance, making them an important target for intervention. However, it is also important to consider other QS systems, such as AI-2 and AIP, which have significant roles in different bacterial species. Future research could expand to target these additional QS pathways, offering new opportunities for the development of broader-spectrum quorum-sensing inhibitors and further insights into microbial communication mechanisms.

## 5. Conclusion

In conclusion, the identification of Sulfaperin and Sulfamerazine as potential quorum sensing inhibitors (QSIs) aligns with drug development efforts to combat antimicrobial resistance. These compounds could serve as lead scaffolds for the development of small-molecule QSIs, which could be optimized through medicinal chemistry approaches such as structure-activity relationship (SAR) studies. Additionally, integrating QSIs into existing antibiotic therapies may enhance treatment efficacy by disrupting bacterial communication and reducing virulence.. Future experimental validation should include in vitro bacterial assays to confirm QS inhibitory activity. Standard quorum sensing inhibition assays, such as the Chromobacterium violaceum CV026 bioassay, can be used to assess AHL degradation or QS disruption. Additionally, measuring QS-regulated virulence factor production (e.g., rhamnolipid, elastase, or pyocyanin production in *P. aeruginosa*) would provide functional validation of the computational findings. Structural characterization through crystallography or surface plasmon resonance could further confirm ligand-protein interactions, while repurposed compounds could expedite the transition to preclinical testing due to their established safety profiles.

## Supporting information

**S1 Fig. Box plot comparing the binding affinities (in kcal/mol) of compounds toward LasI and QscR proteins.** The median binding affinities are represented by red lines within the boxes. While QscR shows a broader distribution of affinities, statistical analysis using the Mann–Whitney U test (U = 104.0, p = 0.800) indicates no significant difference between the two groups (p > 0.05), suggesting comparable binding tendencies of the compounds toward both targets.
(PNG)

**S2 Fig. RMSD analysis of protein-ligand complexes over a 200 ns simulation.** Root Mean Square Deviation (RMSD) analysis for LasI complexed with Quercetin (CID 5325), LasI with Ginkgolide A (CID 68933), QscR with Chloro-N-(4-fluorobenzyl)thiophene-2-sulfonamide (CID 893742), and QscR with N-(carbamoylcarbamothioyl)-2-chlorobenzamide (CID 2796468). Protein RMSD (cyan) and ligand RMSD (red) are plotted, with the X-axis representing time (nanoseconds) and Y-axis for RMSD (nanometers). Results demonstrate stable complexes with minimal fluctuations.
(PNG)

**S3 Fig.** RMSF analysis of protein-ligand complexes over a 200 ns simulation. Root Mean Square Fluctuation (RMSF) analysis highlighting the flexibility of residues in each protein-ligand complex. Specific regions with notable fluctuations are identified, such as residues 100–150 in the LasI-Quercetin complex and residues 25–50 in other complexes, reflecting ligand influence on protein dynamics.
(PNG)

**S4 Fig. SASA analysis of protein-ligand complexes over a 200 ns simulation.** Solvent Accessible Surface Area (SASA) analysis showing differences in solvent exposure among the complexes. LasI-Quercetin exhibited the highest SASA values (130–135 nm²), while QscR-Chloro-N-(4-fluorobenzyl)thiophene-2-sulfonamide displayed lower SASA values (100–110 nm²).
(PNG)

**S5 Fig. Radius of Gyration (Rg) analysis of protein-ligand complexes.** Radius of Gyration (Rg) analysis illustrates the compactness of the complexes. LasI-Quercetin complex remained stable (1.65–1.7 nm), while QscR-N-(carbamoylcarbamothioyl)-2-chlorobenzamide showed wider Rg variations (2.05–2.25 nm), indicating conformational flexibility.
(PNG)

## Consent for Publication

Not applicable.

## Author contributions

**Conceptualization:** Ali alisaac.

**Data curation:** Ali alisaac.

**Formal analysis:** Ali alisaac.

**Funding acquisition:** Ali alisaac.

**Investigation:** Ali alisaac.

**Methodology:** Ali alisaac.

**Project administration:** Ali alisaac.

**Resources:** Ali alisaac.

**Software:** Ali alisaac.

**Supervision:** Ali alisaac.

**Validation:** Ali alisaac.

**Visualization:** Ali alisaac.

**Writing – original draft:** Ali alisaac.

**Writing – review & editing:** Ali alisaac.

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
