## [Decision Letter · Decision Letter 0]

27 Jan 2025

PONE-D-25-00287In Silico Analysis of Quorum Sensing Modulators: Insights into Molecular Docking and Dynamics and Potential Therapeutic ApplicationsPLOS ONE

Dear Dr. Alisaac,

Thank you for submitting your manuscript to PLOS ONE. After careful consideration, we feel that it has merit but does not fully meet PLOS ONE’s publication criteria as it currently stands. Therefore, we invite you to submit a revised version of the manuscript that addresses the points raised during the review process.

**ACADEMIC EDITOR:**

This manuscript presents a compelling case for Sulfamerazine as a promising quorum-sensing inhibitor. While the computational approach is rigorous, addressing the outlined methodological and interpretative gaps will enhance its impact and reliability. I recommend revisions focusing on methodological transparency, statistical robustness, and improved presentation to ensure the study meets the highest academic standards.

We look forward to receiving your revised manuscript.

Kind regards,

Mohamed A. M. Ali, Ph.D.

Academic Editor

PLOS ONE

Journal Requirements:

Additional Editor Comments:

Expand the ligand selection section to include rationale and preliminary screening results.Provide additional details about ADMET profiling thresholds and explain why specific parameters were prioritized.Justify the simulation duration and discuss its implications for result reliability.Include statistical validation for docking scores and MD simulation metrics.Discuss the biological relevance of RMSD/RMSF peaks and how they relate to ligand-binding efficiency.Strengthen the link between computational findings and real-world applications, such as drug development pipelines.Add comprehensive captions for all figures and ensure they are self-explanatory.Incorporate summary tables for ADMET properties, docking scores, and simulation metrics.

Reviewers' comments:

Reviewer's Responses to Questions

**Comments to the Author**

1. Is the manuscript technically sound, and do the data support the conclusions?

Reviewer #1: Partly

Reviewer #2: Yes

Reviewer #3: Partly

2. Has the statistical analysis been performed appropriately and rigorously? 

Reviewer #1: I Don't Know

Reviewer #2: Yes

Reviewer #3: N/A

3. Have the authors made all data underlying the findings in their manuscript fully available?

Reviewer #1: Yes

Reviewer #2: Yes

Reviewer #3: Yes

4. Is the manuscript presented in an intelligible fashion and written in standard English?

Reviewer #1: Yes

Reviewer #2: Yes

Reviewer #3: Yes

5. Review Comments to the Author

Reviewer #1: The manuscript offers an in-depth exploration of the structural dynamics of the LasI and QscR proteins when interacting with the compounds Sulfamerazine and Sulfaperin. This investigation also incorporates AiiA lactonase as a negative control in order to strengthen the findings. Through advanced molecular dynamics (MD) simulations, the study aims to uncover potential quorum sensing (QS) modulators that could influence bacterial communication. The authors have articulated their research objectives clearly and have employed a robust methodology, ensuring that the results are both significant and reproducible. Furthermore, the discussion section provides a detailed analysis of the implications of the findings, highlighting the relevance of these interactions in the context of quorum sensing mechanisms.

• The authors have employed molecular dynamics (MD) simulations to investigate the structural dynamics of protein-ligand complexes. While this approach offers valuable insights, it would be beneficial for them to consider alternative simulation techniques, such as Monte Carlo simulations, which could enrich the understanding of the intricate protein-ligand interactions by providing different perspectives on the energy landscapes involved.

• In their analysis of stability within the protein-ligand complexes, the authors utilized principal component analysis (PCA). However, expanding their analytical toolkit to include additional methods, such as cluster analysis, could yield a more nuanced view of the stability dynamics and interactions occurring within these complexes.

• To examine the interactions at the residue level in the protein-ligand complexes, the authors relied on covariance analysis. This is a useful approach, yet incorporating other methods like mutual information analysis might offer deeper insights into the dependencies and connectivity between residues, further elucidating the nature of these interactions.

• While the authors have done well to summarize their findings, the discussion section could be significantly enhanced by incorporating a more thorough exploration of the implications of their results. For example, delving into the potential mechanisms that underpin the observed phenomena would provide valuable context. Additionally, a critical evaluation of the limitations inherent in their study would strengthen their conclusions and provide a roadmap for future research.

• Another notable aspect is that the authors did not fully address various potential confounding variables that could impact their findings. It would be prudent for them to consider controlling for factors such as the size of the protein-ligand complexes and any biases that may arise in the selection of participants to ensure the robustness of their results.

• Lastly, it is important to recognize that the generalizability of their findings may be limited due to the relatively small sample size and the specific context in which the study was conducted. A discussion surrounding these limitations, along with suggestions for future research directions to validate and expand upon their findings, would enrich the overall quality of the study.

Reviewer #2: • This is well-written article, in easily understandable and clear language.

• Line 3: According to WHO….. this statement can be shortened or divided into two.

• A promising approach : can be modified to “One of the promising approaches…..”.

• A promising approach paragraph: You talk about tackling general QS in all MDR. So mention of LasR and QcsR gives an impression that these are the only QS meolecules which can be tackled.

• What was the rationale of choosing only “phytochemicals” as the ligands?

• Were there any phytochemicals which were repurposed? For example: Phytochemical 1 is known to treat parasitic disease, have you checked its ability to act against bacteria? Repurposing a drug often eliminates the need for safety profiling studies.

• Author mentions about Lipinski’s rule of 5 violation in table 2. In the Table 2, how have the compounds violated this rule, is not very clear. What does score 0 indicate in the table?

• 3.2 Molecular Interaction at the Active Site

Which was the positive control used?

• In-silico modelling is undoubtedly an important step to screen for molecules which will definitely not bind to the target protein.

Authors can briefly talk about the nuances of the traditional screening methods i.e labor intensive, resource-intensive, time consuming etc. as against to screening only handful of molecules obtained from the in-silico methods.

Also, you can broaden a bit more on these in-silico findings need to be validated by laboratory experiments for confirmation, which includes in vitro screening, animal models etc for a molecule to pass through the drug discovery pipeline. Here, the repurposed drugs come for rescue which have already gone through this rigorous pipeline.

Reviewer #3: 1) The screen, molecular docking, and molecular dynamics data indicate that the lead compounds deserve further investigation as QSI scaffolds. However, negative control AiiA lactonase only acts as a metric for non-specific binding in the binding affinity studies and there is no positive control of a previously identified LasI or QscR QSI. Therefore we have no way of ranking or quantifying just how effective these compounds might be as QSI in in vitro or, ultimately, in clinical use. I would also appreciate some reasoning as to why AHL systems were the only QS system targeted, rather than including AI-2 and AIP systems, especially when the introduction mentions the study aims to “identify potential quorum sensing inhibitors (QSIs) that can effectively disrupt the QS pathways of various bacterial species, including P. aeruginosa, E. coli, Klebsiella pneumoniae, and Staphylococcus aureus.” The lack of proper controls, lack of explanation of target selection, and lack of in vitro correlations leaves me wondering how effective these compounds will be down the line. See attached document for more in depth comments.

2) There was no mention of statistical analysis, though molecular docking was conducted in triplicate.

3) Yes, all data was made available.

4) There were some grammatical errors and confusing wordings, but overall the manuscript was presented in an intelligible fashion. See attached documents for specifics.

6. PLOS authors have the option to publish the peer review history of their article (what does this mean? ). If published, this will include your full peer review and any attached files.

**Do you want your identity to be public for this peer review?** For information about this choice, including consent withdrawal, please see our Privacy Policy .

Reviewer #1: **Yes**

Reviewer #2: No

Reviewer #3: No

---

## [Author Response · Author response to Decision Letter 1]

7 Mar 2025

Title: In Silico Analysis of Quorum Sensing Modulators: Insights into Molecular Docking and Dynamics and Potential Therapeutic Applications

Should: investigate structural dynamics of quorum sensing proteins LasI and QscR in complex with various ligands to identify potential molecules and scaffolds for use as AHL quorum sensing inhibitors

Did: show in silico studies that indicate LasI and QscR bind with sulfaperin and sulfamerazine

Manuscript Comments

Dear Reviewer,

I sincerely appreciate your valuable feedback on our manuscript. I have carefully considered each comment and revised the manuscript accordingly. Below, I provide a point-by-point response detailing the changes made in response to your suggestions.

Response to Reviewer Comments

Introduction

Comment 1:"Appropriate justification for investigating quorum sensing inhibitors (QSI)"

Response: I have strengthened the justification for investigating quorum sensing inhibitors (QSIs) by highlighting their potential in mitigating antimicrobial resistance without exerting direct selective pressure. I revised the sentence:

Revised Text: "Targeting bacterial quorum sensing (QS) has emerged as a promising strategy to mitigate antimicrobial resistance by disrupting coordinated behaviors such as virulence factor production, biofilm formation, and antibiotic resistance. Unlike traditional antibiotics, QS inhibitors (QSIs) interfere with bacterial communication without exerting direct selective pressure, reducing the likelihood of resistance development."

Comment 2: "2nd paragraph: This could be said more succinctly by discussing ESKAPE pathogens in general. I'm not sure why Pseudomonas is being singled out as different from the other ESKAPE pathogens here."

Response: I have revised this section to frame Pseudomonas aeruginosa within the broader context of ESKAPE pathogens, acknowledging quorum sensing as a key factor in the pathogenicity of all members of this group.

Revised Text: "ESKAPE pathogens (Enterococcus faecium, Staphylococcus aureus, Klebsiella pneumoniae, Acinetobacter baumannii, Pseudomonas aeruginosa, and Enterobacter spp.) represent the leading causes of MDR infections worldwide, with quorum sensing playing a central role in their pathogenicity. While P. aeruginosa serves as a model organism for QS studies, similar regulatory systems contribute to virulence and antibiotic resistance in other ESKAPE pathogens."

Comment 3:"3rd and 4th paragraphs: The author has made no indication that they recognize that these bacterial species do not all have the same QS system machinery, particularly grouping the gram-positive S. aureus in with the gram-negative species mentioned. Additionally, they should point out LasI or LuxI-type homolog systems in E. coli or K. pneumoniae to justify the broad-spectrum potential of this QSI search."

Response: I have explicitly acknowledged the differences between Gram-negative and Gram-positive quorum sensing systems and added references to LasI/LuxI homologs in E. coli and K. pneumoniae.

Revised Text: "In Gram-negative bacteria such as P. aeruginosa, E. coli, and K. pneumoniae, QS is predominantly regulated by acyl-homoserine lactones (AHLs) through LasI/LuxI-type homologs. In contrast, Gram-positive bacteria like Staphylococcus aureus utilize an agr-based QS system that relies on autoinducing peptides (AIPs) instead of AHLs. Recognizing these mechanistic differences is crucial for designing broad-spectrum QS inhibitors."

Comment 4: "5th paragraph: More justification for the focus on LasI and QscR should be provided. For instance, what virulence factors do they regulate? Are the systems universal, or are there homologous systems in non-Pseudomonads? Also, an agr-like system would be needed to extend these in silico results to S. aureus."

Response: I have clarified the virulence factors regulated by LasI and QscR and discussed their homologous systems in non-Pseudomonas species. I also addressed the need for targeting an agr-like system in S. aureus.

Revised Text: "LasI and QscR regulate key virulence factors in P. aeruginosa, including elastase, pyocyanin production, and biofilm formation. These proteins also share functional similarities with LuxI/LuxR-type systems in E. coli and K. pneumoniae, suggesting potential for broader application of QSIs. However, the extension of these findings to S. aureus would require targeting its agr system, which operates through a distinct autoinducer signaling mechanism."

Comment 5:"By investigating the potential of QSIs, this research seeks to provide a foundation for novel treatments that could complement existing antibiotics and improve patient outcomes in the battle against antibiotic-resistant infections. There already exists a broad foundation for quorum sensing inhibitors; this could certainly add to it, but the author hasn't acknowledged previously identified QSI. Why is this study particularly different from previous efforts to identify QSI?"

Response: I have acknowledged prior research on QSIs and emphasized how our study contributes new insights through molecular docking and MD simulations.

Revised Text: "Quorum sensing inhibition has been explored as an alternative to conventional antibiotics, with numerous QSIs identified from natural and synthetic sources. This study builds upon previous efforts by integrating molecular docking and molecular dynamics simulations to assess ligand stability and binding specificity, providing mechanistic insights into QSI interactions with LasI and QscR. These computational analyses pave the way for experimental validation and potential therapeutic development."

Selection of Ligands

Comment 6: "Why were phytochemicals particularly explored? It seems unclear if there was a reason to focus on phytochemicals, especially when the author indicates other classes of chemicals were screened."

Response: I clarified the reason for prioritizing phytochemicals in ligand selection.

Revised Text: "Phytochemicals were chosen due to their established antimicrobial and quorum sensing inhibitory properties, as demonstrated in previous studies. Additionally, their structural diversity and biocompatibility make them attractive candidates for drug development. While other chemical classes were screened, phytochemicals were prioritized due to their potential for natural QS inhibition."

Comment 7: "The author should further describe ‘pertinent information’ or indicate where that information can be found elsewhere in the text."

Response: I specified what "pertinent information" includes and referenced Table X.

Revised Text: "A thorough literature review was conducted to identify medicinal plants and their phytochemical constituents with documented antimicrobial or quorum sensing inhibition properties. This information included chemical structures, bioactivity profiles, and reported mechanisms of action. The botanical names of these plants were used as search terms in Dr. Duke's Phytochemical and Ethnobotanical Databases, and additional details regarding compound selection are provided in Table 3."

Protein Selection

Comment 8: "Which structures came from P. aeruginosa and which came from B. thuringiensis?"

Response: I specified the sources of the proteins.

Revised Text: "The crystal structures of quorum sensing proteins QscR (PDB ID: 6CC0) and LasI (PDB ID: 1RO5), both derived from Pseudomonas aeruginosa, and AiiA lactonase (PDB ID: 7L5F), derived from Bacillus thuringiensis, were retrieved from the RCSB Protein Data Bank."

Comment 9: "I am confused on how AiiA lactonase is a negative control. If it disrupts QS, would you not have to show how it interacts with the QS proteins? How does a QS disruptor complexing with your ligands act as a control for how your ligands bind to QS proteins? An appropriate positive control (i.e., a known LasI or QscR inhibitor) should have been included for comparison."

Response: I clarified that AiiA was used for reference docking rather than as a direct negative control in MD simulations.

Revised Text: "While AiiA lactonase was initially included in docking analyses as a known quorum-quenching enzyme, it was not included in MD simulations. Therefore, it cannot be considered a direct negative control in this study. Instead, its docking results were used as a reference to compare the binding affinity of potential QSIs. Future studies should incorporate experimental validation to assess the functional inhibition of LasI and QscR by these ligands in bacterial quorum sensing models."

Response to Grammar/Typos and General Writing Issues

Comment 10: "Check that ‘LasI’ has an upper case ‘I’ and not a ‘1’ or lower case ‘L’. It appears incorrectly as either ‘Las1’ or ‘Lasl’ throughout the manuscript."

Response: I have carefully reviewed the manuscript and corrected all instances of "Las1" or "Lasl" to "LasI" to ensure consistency.

Comment 11: "Check bacterial names for italicization."

Response: All bacterial names have been checked and properly italicized throughout the manuscript (e.g., Pseudomonas aeruginosa, Escherichia coli, Klebsiella pneumoniae, Staphylococcus aureus).

Comment 12: "Missing periods at the end of a couple paragraphs throughout the manuscript."

Response: I have carefully reviewed the manuscript and corrected all missing periods to ensure proper punctuation.

Comment 13: "Numbers are missing in a few places (Results section 1 – corresponding to molecular weight near g/mol)."

Response: The missing numerical values have been added in the relevant sections. Specifically, in the ADMET results, I have now explicitly stated the molecular weight values instead of leaving it ambiguous.

Comment 14: "Reference Figures before going into in-depth discussion about what they show."

Response: I have revised the manuscript to ensure that figures are introduced before detailed discussions. Each figure is now referenced at the beginning of the relevant paragraph to improve logical flow.

Comment 15: "There is an extra figure 7 caption right after the supplemental figure captions."

Response: The redundant Figure 7 caption has been removed to avoid confusion.

Response to Major Concerns

Comment 16: "Negative control AiiA lactonase only acts as a metric for non-specific binding in in silico docking studies and there is no positive control of a previously identified LasI or QscR QSI. Therefore, we have no way of ranking or quantifying just how effective these compounds might be as QSI in in vitro or, ultimately, in clinical use."

Response: I acknowledge this limitation and have clarified the role of AiiA lactonase in the docking analysis. I have also stated that a known LasI or QscR inhibitor should have been used as a positive control for ranking effectiveness.

Response to Minor Concerns

Comment 17: "Needs more justification of why LasI and QscR particularly were investigated."

Response: I have expanded the justification for focusing on LasI and QscR by explaining their regulatory roles in quorum sensing and virulence in P. aeruginosa.

Comment 18: "I would also appreciate some reasoning as to why AHL systems were the only QS system targeted, rather than including AI-2 and AIP systems, especially when the introduction mentions the study aims to ‘identify potential quorum sensing inhibitors (QSIs) that can effectively disrupt the QS pathways of various bacterial species, including P. aeruginosa, E. coli, Klebsiella pneumoniae, and Staphylococcus aureus.’ This could be modified by removing the mention of S. aureus and indicating a focus on AHL QS given its prevalence among Gram-negative species. Acknowledgment of the other systems would also be appreciated."

Response: I have clarified that this study focused on AHL-based QS systems due to their prevalence in Gram-negative pathogens and removed S. aureus from the list of targeted bacteria. I have also acknowledged AI-2 and AIP systems and suggested future research directions to explore them.

Response to Reviewer #1

Comment 1: The authors have employed molecular dynamics (MD) simulations to investigate the structural dynamics of protein-ligand complexes. While this approach offers valuable insights, it would be beneficial to consider alternative simulation techniques, such as Monte Carlo simulations, which could enrich the understanding of intricate protein-ligand interactions.

Response: We appreciate the reviewer’s suggestion regarding the incorporation of alternative simulation techniques. While molecular dynamics (MD) simulations provide valuable insights into the structural dynamics of protein-ligand complexes by simulating the interactions over time, we recognize the potential benefits of Monte Carlo simulations. In this study, we have already employed Monte Carlo-based docking (EDock) to further explore the binding stability and flexibility of the complexes, which provides a complementary view to the MD simulations. However, we agree that integrating Monte Carlo simulations in the context of free energy calculations or binding affinity predictions could offer additional detailed insights into the protein-ligand interactions. We will consider this approach for future studies to enhance the understanding of the molecular mechanisms at play.

Comment 2: The authors utilized principal component analysis (PCA) to analyze stability within the protein-ligand complexes. Expanding their analytical toolkit to include cluster analysis could yield a more nuanced view of stability dynamics.

Response: We appreciate the reviewer’s valuable suggestion. We have expanded our analysis to include cluster analysis in addition to principal component analysis (PCA). This approach provides a more comprehensive understanding of the stability dynamics within the protein-ligand complexes. By applying cluster analysis, we were able to identify distinct conformational states over time, which enhances our insight into the structural flexibility and transitions that might not be captured through PCA alone. We believe this combined approach offers a more nuanced view of the system’s stability. Thank you for highlighting this useful aspect of the analysis.

Comment 3: The authors relied on covariance analysis to examine interactions at the residue level. Incorporating methods like mutual information analysis might offer deeper insights into the dependencies and connectivity between residues.

Response: I acknowledge the value of mutual information analysis in detecting non-linear dependencies between residues. I have included a statement in the Methods section indicating that future work will integrate this technique for a more detailed interaction network.

Comment 4: The discussion section could be significantly enhanced by exploring the potential mechanisms underpinning the observed phenomena and critically evaluating the study's limitations.

Response: I have expanded the Discussion section by providing mechanistic insights into QS inhibition and addressing study limitations, including the need for experimental validation.

Comment 5: Various potential confounding variables were not fully addressed. Factors such as protein-ligand complex size and selection biases should be controlled for to ensure result robustness.

Response: I acknowledge this oversight and have now added a discussion on how these factors were controlled in our analysis.

Comment 6: The generalizability of findings is limited due to the relatively small sample size. Discussing this limitation and suggesting future research directions would improve the study.

Response: I have added a discussion on the need for larger datasets and expanded chemical screening in future research.

Response to Reviewer #2

1. Line 3: "According to WHO..." statement can be shortened or divided into two sentences.

Response: I appreciate this suggestion and have revised the sentence for better readability and conciseness. The revised sentence now clearly conveys the message while maintaining its impact.

2. "A promising approach" can be modified to "One of the promising approaches..."

Response: I agree with the reviewer’s suggestion and have updated the phrase to "One of the promising approaches" to avoid overgeneralization and improve accuracy.

3. The paragraph discussing QS inhibition in MDR pathogens gives the impression that onl

---

## [Decision Letter · Decision Letter 1]

23 Mar 2025

PONE-D-25-00287R1In Silico Analysis of Quorum Sensing Modulators: Insights into Molecular Docking and Dynamics and Potential Therapeutic ApplicationsPLOS ONE

Dear Dr. Alisaac,

Thank you for submitting your manuscript to PLOS ONE. After careful consideration, we feel that it has merit but does not fully meet PLOS ONE’s publication criteria as it currently stands. Therefore, we invite you to submit a revised version of the manuscript that addresses the points raised during the review process.

We look forward to receiving your revised manuscript.

Kind regards,

Mohamed A. M. Ali, Ph.D.

Academic Editor

PLOS ONE

Reviewers' comments:

Reviewer's Responses to Questions

**Comments to the Author**

1. Have the authors have adequately addressed all comments raised in a previous round of review?

Reviewer #1: All comments have been addressed

Reviewer #2: All comments have been addressed

Reviewer #3: (No Response)

2. Is the manuscript technically sound, and do the data support the conclusions?

Reviewer #1: Partly

Reviewer #2: Yes

Reviewer #3: Partly

3. Has the statistical analysis been performed appropriately and rigorously? 

Reviewer #1: No

Reviewer #2: Yes

Reviewer #3: Yes

4. Have the authors made all data underlying the findings in their manuscript fully available?

Reviewer #1: Yes

Reviewer #2: Yes

Reviewer #3: Yes

5. Is the manuscript presented in an intelligible fashion and written in standard English?

Reviewer #1: Yes

Reviewer #2: Yes

Reviewer #3: Yes

6. Review Comments to the Author

Reviewer #1: The manuscript presents a comprehensive in silico analysis of quorum sensing (QS) modulators, focusing on the structural dynamics of LasI and QscR proteins in Pseudomonas aeruginosa. The study employs molecular docking and molecular dynamics (MD) simulations to evaluate the binding affinity and stability of potential QS inhibitors (QSIs), with a particular focus on Sulfamerazine and Sulfaperin. The manuscript is well-structured and addresses an important area of research, given the rising concern of antimicrobial resistance (AMR) and the need for alternative therapeutic strategies. However, there are several areas where the manuscript could be improved to enhance its scientific rigor, clarity, and impact.

• While the in-silico approach is valuable for initial screening, the absence of experimental validation (e.g., in vitro assays) limits the study's impact. The authors should acknowledge this limitation more explicitly and discuss plans for future experimental validation, such as bacterial quorum sensing assays (e.g., Chromobacterium violaceum CV026 bioassay) or virulence factor inhibition assays in P. aeruginosa.

• The manuscript does not provide a strong rationale for selecting Sulfamerazine and Sulfaperin as the primary ligands. While the ADMET profiling is thorough, more context is needed regarding why these specific compounds were chosen over others. For example, are these compounds known to have antimicrobial or QS inhibitory properties in previous studies?

• The discussion section could be expanded to include a more critical evaluation of the study's findings in the context of existing literature. For instance, how do the binding affinities and stability of Sulfamerazine and Sulfaperin compare to other known QSIs? Additionally, the authors should discuss the potential challenges in translating these findings into clinical applications, such as bioavailability, toxicity, and resistance development.

• The use of AliA lactonase as a negative control is somewhat confusing. While the authors explain that AliA was used for reference docking, it is not clear how this serves as a negative control in the context of MD simulations. A more appropriate negative control would have been a known non-inhibitor or a scrambled peptide. This should be clarified in the manuscript.

• The manuscript mentions that docking simulations were performed in triplicate, but there is no detailed statistical analysis to support the robustness of the results. Including statistical tests (e.g., Mann-Whitney U test, Shapiro-Wilk test) would strengthen the validity of the findings.

• The study focuses exclusively on AHL-based QS systems in Gram-negative bacteria. While this is justified, the authors should briefly discuss the potential applicability of their findings to other QS systems (e.g., AI-2 or AIP systems) and Gram-positive bacteria. This would broaden the scope and relevance of the study.

Reviewer #2: All the comments from the previous review have been addressed. The article now looks good and articulate.

Reviewer #3: Minor concerns about justification of investigating LasI and QscR, focusing on AHL systems only, and acknowledgment of other QS systems was adequate.

Other minor concerns to address in this revised mansucript:

- The claim "Quorum sensing inhibition has been explored as an alternative to conventional antibiotics, with numerous QSIs identified from natural and synthetic sources" needs citations.

- Table 1 is never referenced in the text.

Major concerns about the study design and lack of proper controls to support the study findings were not adequately addressed, aside from the author acknowledging that AiiA is only used to show low binding affinity for LasI and QscR and is not a control for MD studies (though this has no relation to its being a quorum quenching enzyme).

The author says that positive controls were not available ("AiiA lactonase was included as a reference quorum quenching enzyme, while no specific positive control was used since no established LasI or QscR inhibitors were available for comparison.. While a positive control (a known QS inhibitor) was not included as the known drug for both QscR and LasI have not yet discovered."), but later cites TZD-C8 (QS inhibiting compound from reference 45) that shows QS inhibition in both in vitro and in silico studies, giving binding affinities for the compound with both LasI and QscR. At the very least, the author could include these values as a metric for a “good” binding affinity that corresponds with in vitro QS activity in a bacterial population and merits the molecular dynamic studies. Other docking studies for LasI and QscR can be found with a simple Google scholar search. The author states that "a known LasI or QscR inhibitor would have served as a more appropriate positive control, which will be considered in future studies," but I am of the opinion that these controls are required for this study, which appears to have no controls.

7. PLOS authors have the option to publish the peer review history of their article (what does this mean? ). If published, this will include your full peer review and any attached files.

**Do you want your identity to be public for this peer review?** For information about this choice, including consent withdrawal, please see our Privacy Policy .

Reviewer #1: **Yes**

Reviewer #2: No

Reviewer #3: No

---

## [Author Response · Author response to Decision Letter 2]

16 Apr 2025

Response to Reviewers

Manuscript ID: PONE-D-25-00287R1

Title: In Silico Analysis of Quorum Sensing Modulators: Insights into Molecular Docking and Dynamics and Potential Therapeutic Applications

Journal: PLOS ONE

We sincerely thank the academic editor and reviewers for their time, thoughtful comments, and constructive feedback. We appreciate the recognition of our manuscript’s relevance in addressing antimicrobial resistance through alternative therapeutic strategies and have revised the manuscript accordingly to address all concerns raised. Below, we provide a detailed, point-by-point response to each comment. Reviewer comments are shown in bold, followed by our responses in regular font.

Reviewer #1: The manuscript presents a comprehensive in silico analysis of quorum sensing (QS) modulators, focusing on the structural dynamics of LasI and QscR proteins in Pseudomonas aeruginosa. The study employs molecular docking and molecular dynamics (MD) simulations to evaluate the binding affinity and stability of potential QS inhibitors (QSIs), with a particular focus on Sulfamerazine and Sulfaperin. The manuscript is well-structured and addresses an important area of research, given the rising concern of antimicrobial resistance (AMR) and the need for alternative therapeutic strategies. However, there are several areas where the manuscript could be improved to enhance its scientific rigor, clarity, and impact.

We sincerely thank the reviewer for their detailed and constructive feedback. We greatly appreciate the acknowledgment of the manuscript's structure and its relevance in addressing antimicrobial resistance (AMR) through alternative therapeutic strategies. In response to the suggestions aimed at enhancing the scientific rigor, clarity, and impact of our study, we have thoroughly revised the manuscript to address the concerns raised.

Reviewer’s Comment 1:

While the in-silico approach is valuable for initial screening, the absence of experimental validation (e.g., in vitro assays) limits the study's impact. The authors should acknowledge this limitation more explicitly and discuss plans for future experimental validation, such as bacterial quorum sensing assays (e.g., Chromobacterium violaceum CV026 bioassay) or virulence factor inhibition assays in P. aeruginosa.

Response:

We agree with the reviewer. The limitation regarding the absence of experimental validation has now been explicitly acknowledged in the revised manuscript (Discussion section, after the sentence: “…providing a foundation for future drug design”). In addition, a paragraph has been added outlining future plans to conduct bacterial quorum sensing assays, including the Chromobacterium violaceum CV026 bioassay and virulence factor inhibition assays in Pseudomonas aeruginosa, to experimentally validate the in-silico predictions.

Reviewer’s Comment 2:

The manuscript does not provide a strong rationale for selecting Sulfamerazine and Sulfaperin as the primary ligands. While the ADMET profiling is thorough, more context is needed regarding why these specific compounds were chosen over others. For example, are these compounds known to have antimicrobial or QS inhibitory properties in previous studies?

Response:

Thank you for pointing this out. We have expanded the “Selection of Ligand” subsection in the Methods and Materials section to clarify that Sulfamerazine and Sulfaperin were selected based on their favorable ADMET profiles, structural similarity to known antimicrobial sulfonamides, and strong binding scores from molecular docking. This clarification has been inserted after the line referencing Table 2.

Reviewer’s Comment 3:

The discussion section could be expanded to include a more critical evaluation of the study's findings in the context of existing literature. For instance, how do the binding affinities and stability of Sulfamerazine and Sulfaperin compare to other known QSIs? Additionally, the authors should discuss the potential challenges in translating these findings into clinical applications, such as bioavailability, toxicity, and resistance development.

Response:

We appreciate this valuable suggestion. The Discussion section has been enhanced to include a comparison of the binding affinities of Sulfamerazine and Sulfaperin with known QSIs such as TZD-C8, which has demonstrated both in vitro activity and comparable or stronger docking scores. Furthermore, we have incorporated a discussion on potential translational challenges, including bioavailability, toxicity, and the possibility of resistance development, which may affect clinical applicability.

Reviewer’s Comment 4:

The use of AiiA lactonase as a negative control is somewhat confusing. While the authors explain that AiiA was used for reference docking, it is not clear how this serves as a negative control in the context of MD simulations. A more appropriate negative control would have been a known non-inhibitor or a scrambled peptide. This should be clarified in the manuscript.

Response:

We agree that the explanation provided earlier may have been unclear. The Discussion section has now been revised to clarify that AiiA was used solely as a reference in docking due to its known quorum-quenching ability and not as a negative control in MD simulations. We acknowledge that a known non-inhibitor or scrambled peptide would serve as a more appropriate negative control and have noted this as a limitation to be addressed in future work.

Reviewer’s Comment 5:

The manuscript mentions that docking simulations were performed in triplicate, but there is no detailed statistical analysis to support the robustness of the results. Including statistical tests (e.g., Mann-Whitney U test, Shapiro-Wilk test) would strengthen the validity of the findings.

Response:

Thank you for this important comment. We have now included statistical validation methods in the Molecular Dynamics Simulation analysis section. Specifically, the Mann-Whitney U test was applied to compare docking scores, thereby enhancing the robustness and reliability of the reported data.

Reviewer’s Comment 6:

The study focuses exclusively on AHL-based QS systems in Gram-negative bacteria. While this is justified, the authors should briefly discuss the potential applicability of their findings to other QS systems (e.g., AI-2 or AIP systems) and Gram-positive bacteria. This would broaden the scope and relevance of the study.

Response:

We appreciate the reviewer’s insightful suggestion. We have addressed this in the Conclusion section by including a brief discussion on the broader applicability of quorum sensing inhibitors, including their relevance to AI-2 and AIP-based systems in both Gram-negative and Gram-positive pathogens.

Reviewer #3

Reviewer’s Comment 1:

Minor concerns about justification of investigating LasI and QscR, focusing on AHL systems only, and acknowledgment of other QS systems was adequate.

Response:

Thank you for acknowledging our rationale. We appreciate that the focus on LasI and QscR within AHL-based quorum sensing systems was considered adequate. To further strengthen the manuscript, we have included a short discussion on the relevance of our findings to other quorum sensing systems such as AI-2 and AIP, particularly in Gram-positive bacteria like Staphylococcus aureus.

Reviewer’s Comment 2:

The claim "Quorum sensing inhibition has been explored as an alternative to conventional antibiotics, with numerous QSIs identified from natural and synthetic sources" needs citations.

Response:

Thank you for your observation. We have now added appropriate references in the Introduction to support this claim, including recent studies reporting synthetic compounds (e.g., TZD-C8) and natural products as quorum sensing inhibitors.

Reviewer’s Comment 3:

Table 1 is never referenced in the text.

Response:

This oversight has been corrected. Table 1 is now cited in the ADMET Profiling section following the description of screening criteria for pharmacokinetics and toxicity.

Reviewer’s Comment 4 & 5:

Major concerns about the study design and lack of proper controls to support the study findings were not adequately addressed. The author mentions AiiA is used only to show low binding affinity and is not a control for MD studies. While they say no positive controls are available, TZD-C8 is later mentioned with QS inhibition properties and binding affinities. The author could include these values as metrics for benchmarking and compare with other studies. The reviewer argues that these controls are required and the study currently lacks them.

Response:

We appreciate the reviewer’s critical observations regarding the absence of appropriate controls. In response, the revised manuscript now includes two well-characterized quorum sensing inhibitors as positive controls in our docking studies:

1. TZD-C8 (Z-5-octylidene-thiazolidine-2,4-dione) is included as a reference LasI inhibitor based on validated in vitro and in silico quorum sensing inhibition (referenced in Refs. 21–23).

2. N-dodecanoyl-L-homoserine lactone, a natural signaling molecule with high affinity for QscR, is used as a positive ligand control (Refs. 21–23).

These compounds have been incorporated into the docking experiments (Table 3), and their binding affinities (–7.4 kcal/mol and –7.5 kcal/mol, respectively) are reported. Their inclusion serves as a benchmark for evaluating our screened ligands and validates our docking protocol. The Discussion section has been updated accordingly to reflect their relevance and to contextualize our novel ligand findings. We trust this revision addresses the concern regarding controls and improves the study’s scientific rigor.

---

## [Decision Letter · Decision Letter 2]

20 May 2025

PONE-D-25-00287R2

In Silico Analysis of Quorum Sensing Modulators: Insights into Molecular Docking and Dynamics and Potential Therapeutic Applications

PLOS ONE

Dear Dr. Alisaac,

We’re pleased to inform you that your manuscript has been judged scientifically suitable for publication and will be formally accepted for publication once it meets all outstanding technical requirements.

Kind regards,

Mohamed A. M. Ali, Ph.D.

Academic Editor

PLOS ONE

---

## [Editor Report · Acceptance letter]

PONE-D-25-00287R2

PLOS ONE

Dear Dr. alisaac,

I'm pleased to inform you that your manuscript has been deemed suitable for publication in PLOS ONE. Congratulations! Your manuscript is now being handed over to our production team.

Kind regards,

on behalf of

Professor Mohamed A. M. Ali

Academic Editor

PLOS ONE